# Focal seizures induce spatiotemporally organized spiking activity in the human cortex

Joshua M. Diamond[1], Julio I. Chapeton [1], Weizhen Xie [1,2], Samantha N. Jackson [1], Sara K. Inati [3] & Kareem A. Zaghloul [1] ✉

Epileptic seizures are debilitating because of the clinical symptoms they produce. These symptoms, in turn, may stem directly from disruptions in neural coding. Recent evidence has suggested that the specific temporal order, or sequence, of spiking across a population of cortical neurons may encode information. Here, we investigate how seizures disrupt neuronal spiking sequences in the human brain by recording multi-unit activity from the cerebral cortex in five male participants undergoing monitoring for seizures. We find that pathological discharges during seizures are associated with bursts of spiking activity across a population of cortical neurons. These bursts are organized into highly consistent and stereotyped temporal sequences. As the seizure evolves, spiking sequences diverge from the sequences observed at baseline and become more spatially organized. The direction of this spatial organization matches the direction of the ictal discharges, which spread over the cortex as traveling waves. Our data therefore suggest that seizures can entrain cortical spiking sequences by changing the spatial organization of neuronal firing, providing a possible mechanism by which seizures create symptoms.

Seizures are pathological events characterized by elevated neuronal excitability and synchronization[1,2]. In focal seizures, abnormal activity generally begins in a relatively confined region of the cortex, and then propagates through gray and white matter pathways to recruit other regions[3–8]. Clinical symptoms often reflect the involvement of particular brain regions in seizures, and therefore semiology, or the study of seizure symptoms and their evolution, has enabled clinicians to identify where seizures originate and propagate in individual patients with epilepsy[9,10]. A key assumption underlying this approach is that the pathological activity characteristic of seizures fundamentally disrupts normal neural coding, thereby producing symptoms.

Decades of research on neural coding have demonstrated that information during normal cognition can be represented by changes in firing rate in neuronal populations[11–13]. Because seizures modify and often elevate spiking rates among cortical neurons[3,14], seizures may disrupt neural coding by limiting the ability of these neurons to modulate their firing rates flexibly. Recent evidence, however, has suggested that the temporal order, or sequence, of firing among a population of neurons may also play a fundamental role in information processing in the human brain[15–18]. Such sequence coding could add efficiency to rate-based codes by assigning meaning to the latency and ordering of action potentials, information that is lost in a strictly rate-based paradigm[18,19]. Hence, neural coding likely relies on the ability to generate variable patterns of spiking activity through changes in spiking rates, spiking sequences, or both. Seizures likely disrupt neural coding, but the precise effect of seizures on sequences of neuronal spiking activity in the human brain remains unclear.

Here, we explore this question by recording populations of neurons in the human cerebral cortex in five male participants with drug-resistant epilepsy. Alongside standard intracranial electrodes used to

[1]Surgical Neurology Branch, NINDS, National Institutes of Health, Bethesda, MD 20892, USA. [2]Department of Psychology, University of Maryland, College Park, MD 20742, USA. [3]Clinical Epilepsy Section, NINDS, National Institutes of Health, Bethesda, MD 20892, USA. ✉e-mail: kareem.zaghloul@nih.gov

monitor seizures, we implanted micro-electrode arrays (MEAs) to better understand how seizures alter sequences of spiking activity among populations of cortical neurons. We find that, during seizures, pathological discharges are associated with bursts of neuronal spiking organized into temporal sequences across the population. These sequences are highly consistent within seizures and diverge from the spiking sequences observed at baseline. In light of recent research suggesting that pathological discharges propagate across the brain as traveling waves[5,7,20], we examine the spatial organization of the spiking sequences. As spiking sequences evolve over the course of the seizure, they become spatially organized so as to match the direction of pathologic traveling waves. Together, our findings suggest that seizures entrain neuronal spiking sequences in the human cortex, changing the spatial organization of neuronal firing. Changes in native neuronal spiking sequences may, therefore, represent a complementary mechanism by which seizures disrupt neural coding and, by consequence, impair normal human brain function.

## Results

### Spiking sequences during pathologic discharges are consistent within and across seizures

We recorded simultaneous intracranial EEG (iEEG), multi-unit spiking activity (MUA), and micro-scale local field potential (LFP) signals in five participants ($27.80 \pm 5.76$ years old, all male; Supplementary Table 1) with drug-resistant epilepsy following a surgical procedure in which we implanted intracranial electrodes for seizure monitoring. We captured MUA and LFP recordings using MEAs placed in cortical regions suspected to be involved in seizures (see Methods). In one of these participants, we placed two MEAs. Because these MEAs record spiking activity from different populations of neurons, we considered each of these MEAs as independent for purposes of analysis. Thus, the data we present here are captured from five participants with a total of six MEAs.

We found that, following recruitment of the MEA to seizures (Supplementary Table 2), ictal discharges observed in the LFP were universally associated with bursts of spiking activity across the MEA (Fig. 1a). We likewise found large bursts of spiking activity associated with many interictal epileptiform discharges (IEDs; see Supplementary Table 3). Consistent with previous studies[17,21], we also found spontaneous bursts of spiking across the MEA during baseline periods between seizures, which were not associated with clear discharges in the LFP. We were interested in comparing bursts arising in different states to each other. Therefore, we detected all bursts of MUA in every array during every recording and designated each burst as a seizure, IED, or baseline burst (see Methods; Supplementary Fig. 1; Supplementary Table 3). Across all participants and epochs, seizure bursts were associated with the highest spike rate, while IED and baseline bursts were associated with progressively lower spike rates, respectively (Supplementary Fig. 2).

In each burst, spiking activity across the MEA is not perfectly synchronized. Instead, the time of peak spiking rate exhibits a temporal order, or sequence, across the micro-electrodes. Visual inspection suggests that this temporal sequence is consistent from one seizure burst to the next (Fig. 1a). We used Spearman's rank correlation to measure the similarity of the sequence of spiking activity in each burst to sequences in other bursts (using Fisher $z$-transformed $\rho$, see Methods). We found strong self-similarity among all seizure bursts in an example seizure from a single participant and array (Fig. 1b, c). Self-similarity was weaker among IED and baseline bursts in this example array.

We were interested in examining the similarity of sequences of spiking activity across bursts from different seizures within the same participant. In this example participant, we captured six recruited seizures spanning multiple days. We compared all bursts to other bursts within the same and across different seizures (Fig. 1d). Mean

similarity across different seizures was about as strong as mean similarity within seizures ($\rho = 0.47 \pm 0.06$ within versus $0.42 \pm 0.02$ across seizures). We computed the mean similarity of burst sequences within and across seizures in each participant and array (in one participant, we only captured one seizure; see Supplementary Table 2). Across participants, similarity among bursts within seizures was comparable to similarity across different seizures ($t(4) = 0.569$, $p = 0.60$, paired $t$-test; Fig. 1e), suggesting that the sequences of spiking activity observed during seizures are consistent even across multiple days.

We then compared the extent to which sequences of spiking activity are similar or different across different types of bursts. Across participants, seizure bursts demonstrated the greatest self-similarity of spiking sequences on average, while IED and baseline bursts exhibited weaker self-similarity (Fig. 1f). Sequence similarity was also weak in comparisons between burst states (seizure versus IED, etc.). Spike sequence similarity among seizure bursts was significantly greater than similarity in any other comparison, including comparisons within and across groups ($F(5, 20) = 23.57$, partial $\eta^2 = 0.84$, $p < 0.001$, repeated measures one-way ANOVA). These data, therefore, suggest that seizure bursts exhibit constrained and stereotyped sequences of spiking activity that are significantly different from the sequences of spiking activity observed at rest or during IEDs.

To visualize and characterize the similarity in spiking sequences between bursts, we projected the sequences of spiking activity from all seizure, IED, and baseline bursts onto a low-dimensional manifold using PCA[22,23] (Fig. 2a; see Methods). In light of their self-similarity, we reasoned that seizure bursts would reside in a distinct subspace, since distances in low-dimensional space tend to track with similarity between sequences (Supplementary Fig. 3). In the same example participant, we found, in fact, that seizure bursts are displaced with respect to baseline and IED bursts, and reside in a particular sub-region of the low-dimensional manifold.

We repeated this low-dimensional projection in all participants and arrays (see Supplementary Figures). In each participant, we first confirmed that the baseline sequences are themselves not simply random sequences (Supplementary Fig. 4) and then identified the centroid of these baseline burst sequences in low-dimensional space. Across participants, there was a significant influence of burst state on distance from the baseline centroid ($F(2, 8) = 119.701$, partial $\eta^2 = 0.97$, $p < 0.001$, repeated measures one-way ANOVA; Fig. 2b). Seizure bursts were significantly displaced from baseline bursts and from IED bursts ($p < 0.001$, Holm-Bonferroni correction for multiple comparisons), although IED bursts were not significantly displaced from baseline bursts ($p = 0.058$). We examined distance from the baseline centroid of baseline and IED burst in the immediate pre- and post-ictal states, and did not find evidence of abnormal pre- or post-ictal entrainment (Supplementary Fig. 5). We confirmed that these differences between seizure, IED, and baseline bursts were not a consequence of the higher spike rates observed during the seizure bursts (Supplementary Fig. 6). We also confirmed that these differences were present even after excluding the participant with a parietal MEA and focusing only on temporal lobe regions (Supplementary Fig. 7).

An advantage of projecting spiking sequences onto a low-dimensional manifold is that it enables us to visualize the evolution of the seizure in sequence space over time. In this example seizure, for instance, seizure bursts are dynamic, evolving along a particular trajectory over the course of the event (Fig. 2a). We characterized this evolution by measuring the Euclidean distance in PCA space between every seizure burst and the baseline centroid (Fig. 2c; distances normalized to maximum possible pairwise distance in PCA space). Seizure bursts begin near the baseline centroid, but shortly after the first seizure burst, abruptly diverge from baseline.

We repeated this analysis for all seizures in all participants and arrays. The average event duration was $243.51 \pm 217.06$ seconds per seizure. To capture seizure dynamics over this timescale while

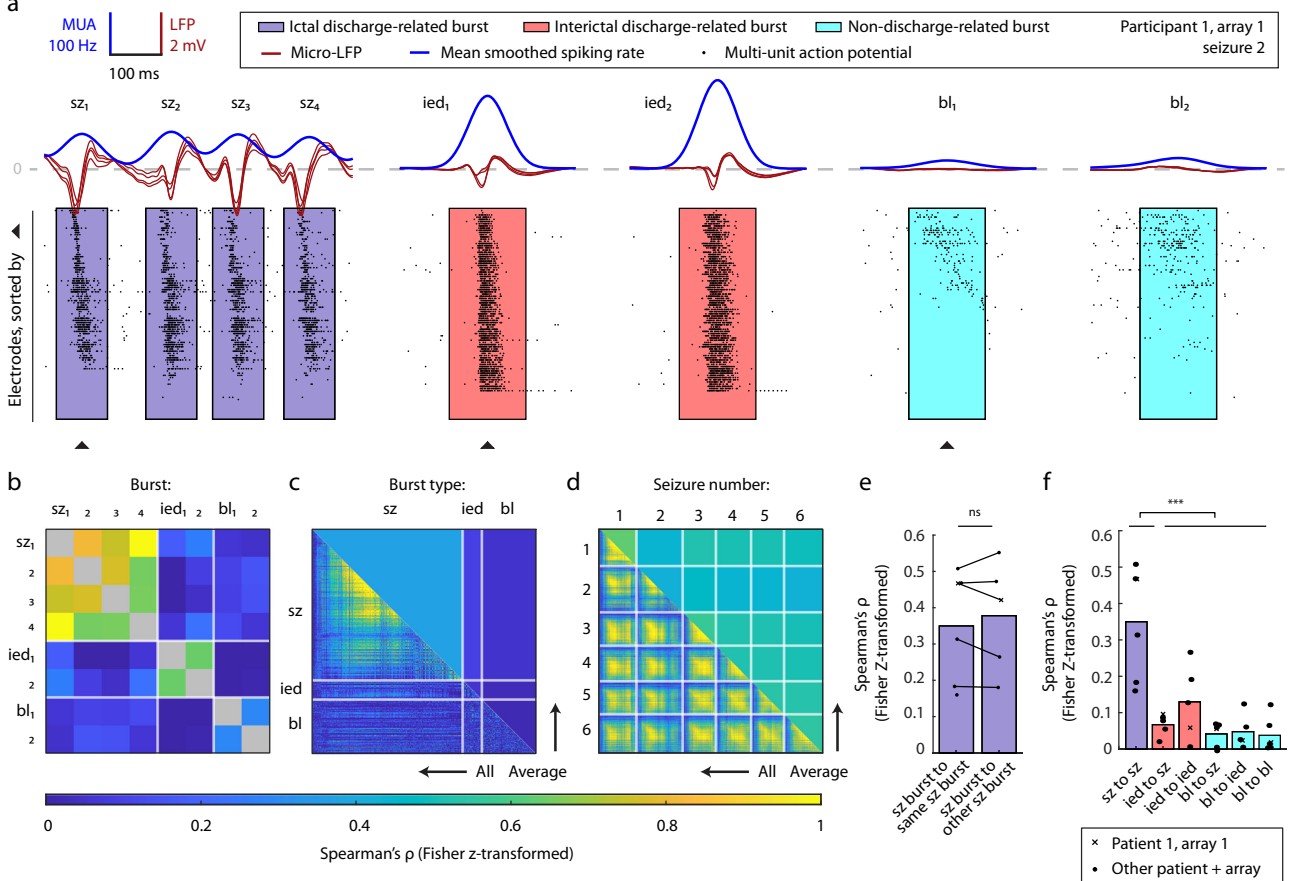

**Fig. 1 | Spiking sequences during pathologic discharges are consistent within and across seizures. a** Raster of multi-unit activity (MUA, black dots) along with simultaneous micro-LFP recordings from four randomly selected micro-electrodes (maroon traces) in one example MEA. We identified spike bursts during seizures (purple), IEDs (red), and during the baseline state (cyan) based on the mean smoothed firing rate across electrodes (dark blue traces; see Methods). The four seizure bursts, sorted according to the order of firing in the first displayed seizure burst ($sz_1$, black triangle), appear to have similar sequences. Raster plot rows in the IED and baseline bursts are sorted based on the order of firing during the first IED and baseline burst, respectively (black triangles). **b** Spearman's rank correlation coefficient (Fisher z-transformed $\rho$, see Methods) comparing sequences of multi-unit activity to each other for the eight bursts shown in **a**. There is strong similarity within seizure bursts, but weaker similarity for other comparisons. **c** Comparison of spike sequences from all detected bursts to one another in this participant and seizure (lower triangle; average of all comparisons in upper triangle). **d** Comparison

of spike sequences from all seizure bursts to one another across all seizures for this participant. **e** Average similarity between spike sequences from all seizure bursts within the same seizures and across different seizures in all participants ($\rho = 0.35 \pm 0.15$ within versus $0.38 \pm 0.15$ across seizures). Each point represents one participant and array (one participant only had one recorded seizure). There is no significant difference between the two groups ($p = 0.60$, two-tailed paired $t$ test). **f** Average similarity within and between seizure (sz), IED, and baseline (bl) bursts for all participants and arrays (self-similarity $\rho = 0.35 \pm 0.15$ for seizure bursts, $0.13 \pm 0.10$ for IED bursts, $0.04 \pm 0.05$ for baseline bursts; similarity is $0.07 \pm 0.03$ for IED versus seizure, $0.04 \pm 0.03$ for baseline versus seizure, and $0.05 \pm 0.05$ for baseline versus IED). Each point represents one participant and array. Spike sequence similarity is significantly greater among seizure bursts than in other comparisons (repeated measures one-way ANOVA; $p < 0.001$, Holm–Bonferroni correction for multiple comparisons).

reducing the influence of noise, we computed the distance between seizure bursts and the baseline centroid averaged over three-second sliding windows (Fig. 2c). From this moving average, we identified the initial, final, maximum, and minimum distances between the seizure bursts and the baseline centroid (Fig. 2d; initial distance $0.16 \pm 0.04$, maximum distance $0.52 \pm 0.08$, minimum distance $0.13 \pm 0.04$, final distance $0.34 \pm 0.07$). Because, by definition, the maximum (minimum) distances will necessarily be greater (less) than or equal to the initial and final distances, we compared these measures of distance to the distances expected by chance. We created random null versions of each seizure burst sequence and computed the distance between random seizure bursts and the baseline centroids (see Supplementary Fig. 8). In these surrogate data, the distance from random null seizure bursts to the baseline centroid, distance from IEDs to the baseline centroid, and distance from baseline bursts to the baseline centroid were not significantly different from each other ($F_{(2, 8)} = 4.191$, partial $\eta^2 = 0.512$, repeated measures ANOVA; $p = 0.11$, Greenhouse-Geisser

correction for non-sphericity). Furthermore, there was no clear evolution of seizure burst location over the duration of the seizure. We therefore normalized the true distances between seizure bursts and baseline centroid, in each participant, by subtracting the initial, final, maximum, and minimum distances in the surrogate data from the true data (Fig. 2e). After this correction, normalized initial distance was $-0.04 \pm 0.09$, maximum distance $0.24 \pm 0.11$, minimum distance $0.02 \pm 0.05$, and final distance $0.18 \pm 0.09$. Across participants, normalized distance between the initial seizure bursts and the baseline centroid was not significantly different from zero ($-0.04 \pm 0.09$; $t(5) = 1.21$; Cohen's $d = -0.42$; $p = 0.28$, one-sample $t$ test). However, as seizures progress, the seizure bursts reach a maximum distance from the baseline centroid that, after normalizing, is significantly greater than zero ($0.24 \pm 0.11$; $t(5) = 5.55$; Cohen's $d = 1.91$; $p = 0.003$, one-sample $t$ test).

These data demonstrate that across participants, the sequences of spiking activity observed during seizure bursts are significantly more

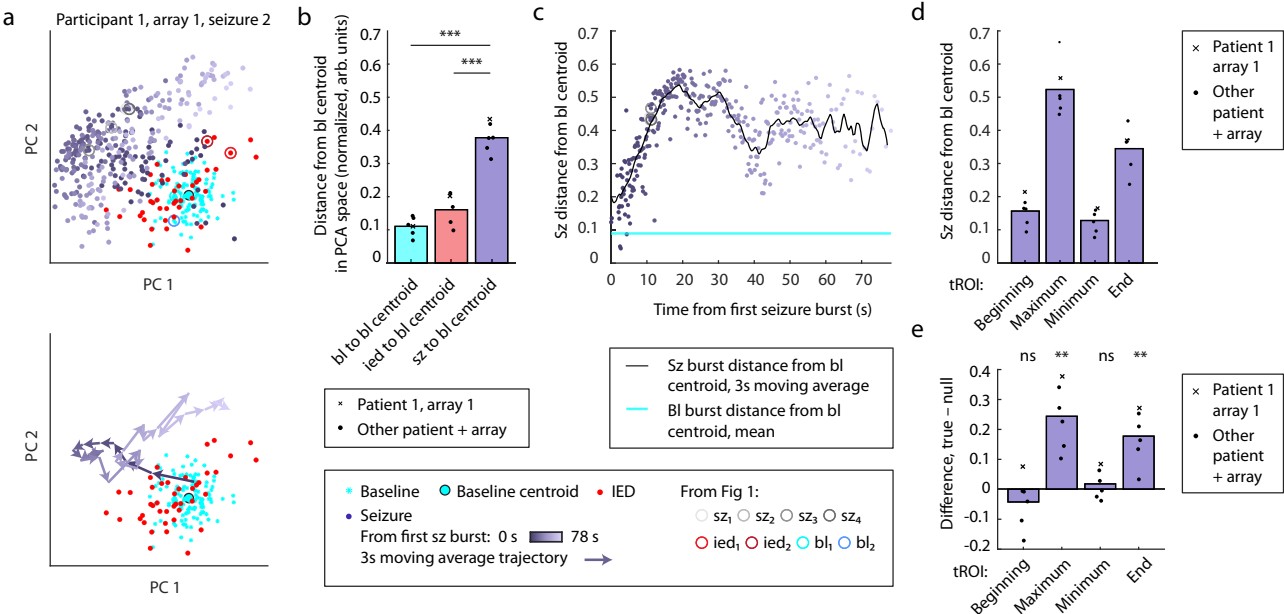

**Fig. 2 | Seizures recruit spiking bursts to a distinct subspace. a** Top: projection of spike sequences onto two-dimensional PCA space in the same example participant. Each dot represents a single spike sequence during a baseline burst (cyan), IED burst (red), and seizure burst (colored dark purple to light purple based on the time from the first detected seizure burst). The four seizure, two IED, and two baseline bursts from Fig. 1 are highlighted (gray, red, and blue rings). We identified the centroid of the baseline bursts in PCA space (cyan circle). Bottom: Identical projection of spike sequences demonstrating the progression of seizure bursts over time. Each arrow reflects the mean location in PCA space for seizure bursts over two consecutive three-second bins. **b** Average distance between seizure (sz) bursts and baseline centroid (purple bar, $0.38 \pm 0.04$, normalized arbitrary units), between IED bursts and baseline centroid (red bar, $0.16 \pm 0.05$), and between baseline (bl) bursts and baseline centroid (cyan bar, $0.11 \pm 0.03$) across participants and arrays. Seizure bursts are significantly farther from the baseline centroid than IED bursts or baseline bursts are (repeated measures one-way ANOVA; two-tailed $p < 0.001$,

Holm-Bonferroni correction for multiple comparisons). **c** Normalized Euclidean distance in PCA space between seizure bursts and the baseline centroid over the course of this example seizure (three-second moving average in black; example seizure bursts from Fig. 1 highlighted with gray rings). Seizure bursts initially reside close to baseline bursts (cyan line indicates mean baseline distance to baseline centroid), but then abruptly diverge. **d** Initial, maximum, minimum, and final distances between seizure bursts and baseline centroids in each participant and array. **e** Initial, maximum, minimum, and final distances between seizure bursts and baseline centroids, normalized by the distances observed after shuffling the sequences of the seizure bursts, in each participant and array (see Supplementary Fig. 8). Seizure bursts travel farther from the baseline centroid at maximum ($p = 0.003$, two-tailed one-sample $t$ test, $p$ value unadjusted) and terminate farther away from the baseline centroid ($p < 0.004$) than would be expected by chance alone.

self-similar and consistent than other sequences, and tend to diverge shortly after seizure onset from the sequences observed at baseline. We were therefore interested in examining whether the divergence in sequences during seizures may be related to their self-similarity. To assess this, we computed Spearman's rank correlation between the spiking sequence of every seizure burst and its eight nearest temporal neighbors (Fig. 3a) to generate a temporally-evolving measure of self-similarity. Self-similarity in this example seizure increases abruptly after the first seizure burst (Fig. 3b). This increase in self-similarity is highly correlated with the increase in distance in low-dimensional space between seizure sequences and the baseline centroid ($\rho = 0.70$; Fig. 3c). We examined this relationship aggregated across all seizures in all participants (Fig. 3d; see Methods). Across all arrays, there was a significant correlation between seizure sequence self-similarity, on the one hand, and the distance between seizure sequences and baseline activity, on the other (mean $\rho = 0.76$, 95% CI 0.28 to 1.24; $t(5) = 4.09$; Cohen's $d = 1.40$; $p = 0.009$, one-sample $t$ test). Together, these data suggest that as seizure sequences diverge from baseline, they become more internally consistent. This, in turn, may suggest that spiking sequences are entrained during seizures so as to exhibit certain stereotyped, characteristic features.

## Seizure bursts become directional as they diverge from baseline bursts

Because neurons and the micro-electrodes from which we record MUA are distributed across spatial locations in the cortex, a spiking sequence with a given temporal order of neurons also reflects a spatial

order, by its very nature. Thus, a sequence of spikes in time could also exhibit a consistent direction in space. On the other hand, if subsequent spikes in a sequence arise from neurons in random locations, then the spike sequence may not exhibit a clear spatial organization.

We, therefore, were interested in examining whether the sequences of spiking activity observed during seizure, IED, and baseline bursts also exhibited spatial organization across the MEA. We mapped the timing of peak MUA activity to the spatial location of each micro-electrode in the MEA during each burst. In the same four example seizure bursts as shown previously, the spike sequences appear to have a degree of spatial organization (Fig. 4a). We used linear regression to fit a plane to the timings of MUA in each sequence (see *Methods*). For every burst, we thereby generated a measure of goodness of fit, $R^2$, which is bounded between 0 and 1, and quantifies the extent to which the spatial order of the sequence is directional. In this example patient, the mean directionality, $R^2$, for both seizure and IED bursts was greater than the mean $R^2$ for random null samples obtained by scrambling the ordering of spiking in seizure bursts (Fig. 4b). We repeated this analysis for all participants and arrays. Across participants and arrays, there was a significant effect of state on $R^2$ ($F(3, 12) = 8.72$, partial $\eta^2 = 0.69$, $p = 0.002$, repeated measures one-way ANOVA; Fig. 4c). There was a statistically significant difference between seizure and null $R^2$ ($p = 0.002$) and between IED and null $R^2$ ($p = 0.02$), but not between baseline and null $R^2$ ($p = 0.13$).

We then asked whether the directional nature of the seizure bursts may be related to their divergence from the baseline on the low-

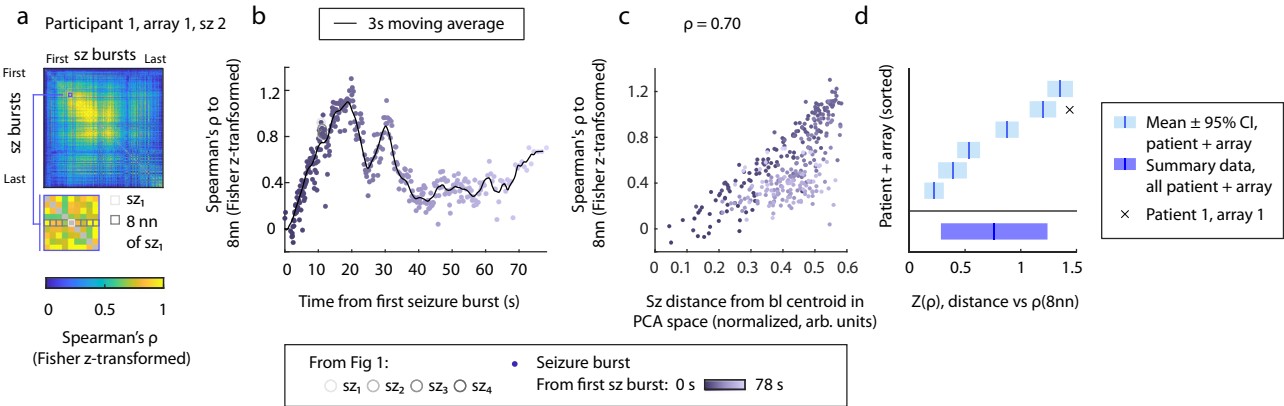

**Fig. 3 | Seizures induce stereotypy among spiking bursts. a** Spearman's rank correlation (Fisher $z$-transformed $\rho$) between spiking sequences of every seizure burst in this same example seizure. To extract a time-evolving measure of spike sequence consistency, we computed the correlation between each spiking burst and its eight nearest temporal neighbors (8nn; see inset). **b** Timecourse of similarity of each seizure burst to eight nearest temporal neighbors in this example seizure (three-second moving average in black). The example seizure bursts from Fig. 1 are highlighted with gray rings. **c** There is a strong correlation between distance from baseline centroid (Fig. 2c) and time-evolving self-similarity of seizure bursts for this example seizure ($\rho = 0.70$). **d** Average correlations between distance from baseline centroid and time-evolving self-similarity for seizure bursts across all seizures in all participants, for $n = 6$ arrays. For each array in each participant, the weighted average of the correlation is shown (dark blue) along with the 95% confidence interval (light blue). The average correlation across all participants and arrays is shown below (dark blue vertical line and error bar). × indicates participant 1.

dimensional manifold. To test this, we examined the evolution over time of the directional nature of the spiking sequences in the same seizure (Fig. 4d). The evolution of directionality over time for this seizure appears similar to the evolution of the distance between seizure sequences and the baseline centroid (Fig. 2c). We found a positive correlation between spatial directionality, as measured by the goodness of fit $R^2$, and the distance from seizure sequences to the baseline centroid in PCA space ($\rho = 0.60$, Fig. 4e). We repeated this analysis for all seizure bursts aggregated across all participants and arrays (Fig. 4f; Supplementary Fig. 9; see Methods). Across participants and arrays, there was a significant and positive relationship between the directionality of the sequences during seizure bursts and their distance from the baseline centroid (mean $\rho = 0.41$; 95% CI 0.09 to 0.72; $t(5) = 3.31$; Cohen's $d = 1.14$; $p = 0.021$, one-sample $t$ test), suggesting that seizure burst sequences become more directional as they diverge from baseline.

**Direction of seizure spiking bursts evolves to reflect the direction of LFP discharges during seizures**

We were interested in determining whether the directional nature of the spiking sequences observed during seizure bursts may be related to the directional nature of the associated ictal discharges. Previous work has demonstrated that ictal and interictal discharges travel over the cortical surface as traveling waves in the LFP[4,5,7,20,24,25]. We, therefore, asked whether LFP traveling waves during seizures influence the directionality of the observed spike sequences.

To examine this, we mapped the timing of the LFP discharges to the spatial location of each micro-electrode in the MEA during each burst (Fig. 5a). Much like the MUA peak timings, the LFP discharges in the same four example seizure bursts have directional spatial organization. We again used linear regression to fit a plane to this spatial organization. The resulting measure of directionality, $R^2$, was larger for the LFP discharges across all bursts in this seizure than for the MUA sequences from the same bursts (Fig. 5b). This difference was consistent across all participants (Fig. 5c). Across all participants and arrays, the LFP discharges exhibited a substantially stronger directional spatial organization, as reflected by a better fit to the plane. The increased spatial organization in the LFP relative to the MUA cannot be attributed simply to increased noise in the MUA (Supplementary Fig. 10).

We then examined how the directionality of the spiking and LFP discharge sequences evolves over the time course of a seizure. We divided each seizure into thirds and separately analyzed the bursts of spiking activity and LFP discharges in each third. In the same example seizure used in previous figures, we extracted the directionality measure $R^2$, reflecting how well the spike sequences and LFP discharges fit with a plane, and the actual direction of that plane, for every burst in every third of the seizure (Fig. 6a, c, e). From the distributions of directions across all bursts, we can see that LFP direction is relatively stable throughout the seizure. MUA direction, on the other hand, changes substantively only in the final third of the seizure, at which time it more closely approximates the direction of the LFP. We explored this directly by computing the angular difference between the estimated spatial directions of the LFP and MUA sequences in each burst, and examining the distribution of these differences across all bursts in each third of the seizure (Fig. 6b, d, f). These angular differences reveal that, in the final third of the seizure, the direction of the spike sequences evolves to match the direction of the LFP discharges.

We examined this evolution in this seizure in more detail by visualizing the spatial direction of the spike and LFP sequences in each individual burst over the course of the entire seizure (Fig. 6g). Although every sequence of spiking activity of LFP discharges can yield some fit to a plane, here we limited our analyses only to those bursts with spatial linear regressions yielding $p \le 0.05$ (60.95% of MUA sequences; 80.18% of LFP sequences; see Methods). For most of the seizure, LFP discharges exhibit a spatial direction in the antero-superior direction, while spike sequences are oriented postero-inferiorly. Towards the end of the seizure, however, around 60 seconds following the initial seizure burst, the spiking sequences dramatically change their spatial direction to align with the LFP discharges in the anterosuperior direction.

We repeated this analysis in all seizures in all participants and arrays, tracking the directional nature of the spike sequences and LFP discharges in each third of each seizure as well as during the interictal discharges (Supplementary Fig. 11; see Supplementary Figs. for all seizures). In every burst, we performed spatial linear regression on both the spike and LFP discharge sequences, and extracted the difference in direction of each fit only in cases when both fits were significant. We then extracted the mode of the distribution of these

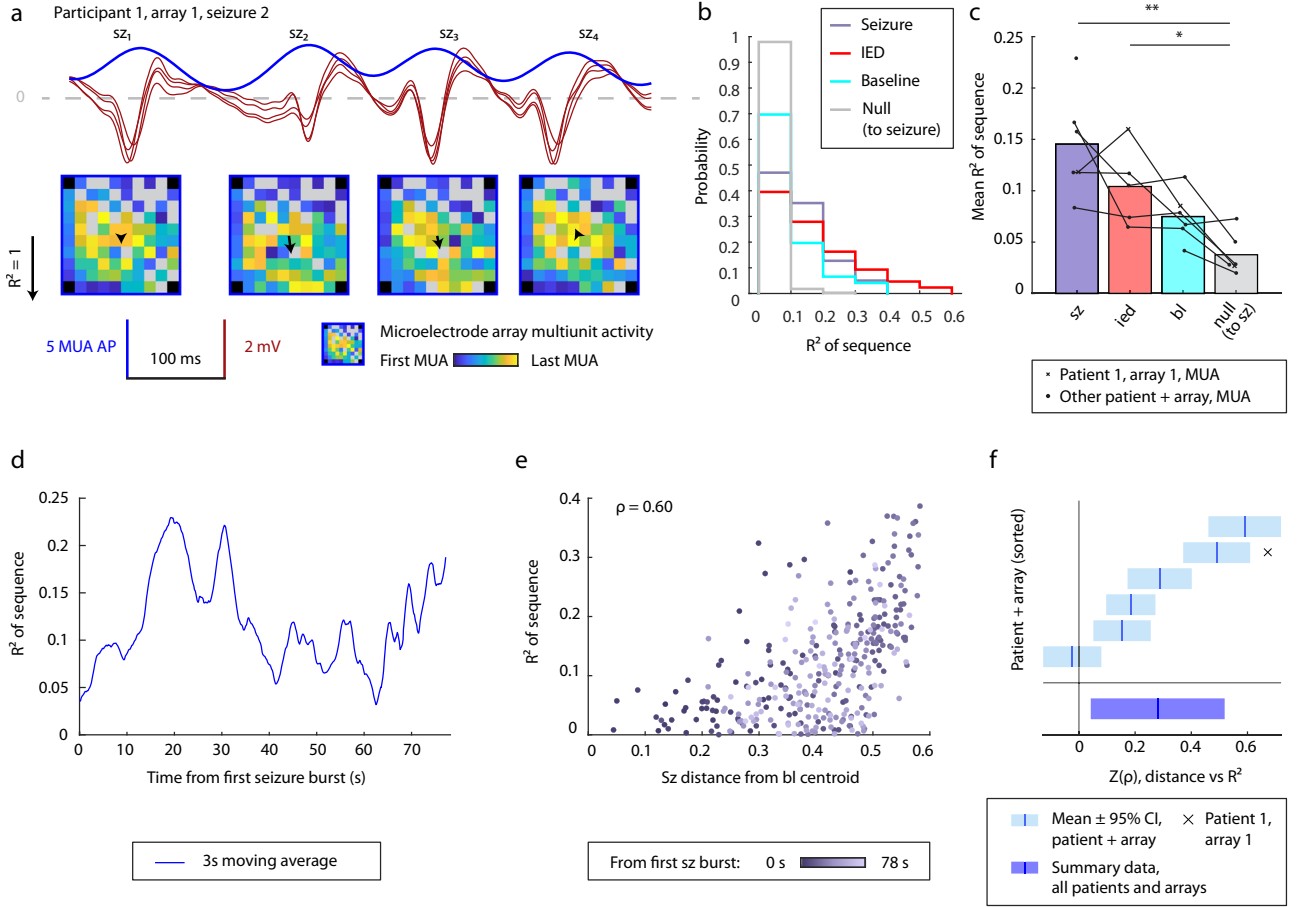

**Fig. 4 | Seizure bursts become directional as they diverge from baseline bursts.** **a** For each of the same four example seizure bursts (mean population spiking rate in dark blue trace; micro-LFP in maroon traces; copied from Fig. 1a), we mapped the timing of peak MUA activity onto the spatial layout of the MEA. Each microelectrode is colored according to the time of peak MUA activity during that seizure burst (blue: early MUA; yellow: late MUA). We fit a plane to the spatial organization of each spiking sequence, generating a measure of goodness of fit ($R^2$, indicated by length of black arrow) and a direction (indicated by direction of black arrow). **b** Distribution of $R^2$ values for all bursts in the same example seizure (purple, mean $0.12 \pm 0.09$) compared to IED bursts (red, mean $0.16 \pm 0.14$), baseline bursts (cyan, mean $0.09 \pm 0.08$), and null bursts in which the order of spiking electrodes were scrambled (gray, $0.03 \pm 0.03$; see Methods). **c** Average $R^2$ over all seizure (sz; $0.15 \pm 0.05$), IED ($0.10 \pm 0.04$), baseline (bl; $0.07 \pm 0.02$), and null bursts

($0.04 \pm 0.02$) in every participant and array. Each dot represents a single participant and array. $R^2$ for seizure bursts and IED bursts was significantly greater than for null ($p = 0.002$ and $p = 0.02$, respectively; two-tailed p-values adjusted using the Holm–Bonferroni correction for multiple comparisons). **d** Timecourse of the goodness of fit ($R^2$) for all seizure bursts in this example seizure (three-second moving average). **e** Correlation between distance from baseline centroid and $R^2$ for each seizure burst ($\rho = 0.60$). **f** Weighted average correlations between distance from baseline centroid and $R^2$, for all seizure bursts across all seizures in each participant, for $n = 6$ arrays. For each array in each participant, the average correlation is shown (*dark blue*) with the 95% confidence interval (light blue). Average correlation across all participants and arrays is shown below (*dark blue* vertical line and error bar). × indicates participant 1.

angular differences in each third of each seizure, and computed the average of these modes across all seizures for each participant and array. On average, in the first third of seizures, the angular difference between the spike and LFP sequences is evenly distributed (Fig. 7a). In the middle third, for several participants, the angle of the spiking sequences during seizures has shifted towards the direction of the LFP discharges. By the final third of seizures, by and large, the direction of spiking sequences matches the direction of the LFP discharges.

We examined this temporal evolution in all participants in more detail by visualizing the difference in spatial directions of the spike and LFP sequences over the course of all seizures (Fig. 7b). We divided each seizure into three-second time bins, and in each bin, computed the angular difference between spike and LFP sequences only for bursts that had a significant spatial fit. Across all seizures, the mean of these angular differences evolves toward zero, suggesting that the direction of the spiking sequences becomes more similar to the direction of the LFP discharges as the seizure progresses.

## Discussion

Our data demonstrate that pathological discharges during seizures are associated with bursts of spiking activity across a population of cortical neurons. Within these bursts, MUA is organized into highly consistent and stereotyped temporal sequences. As the seizure evolves, spiking sequences diverge from the sequences observed at baseline and during interictal discharges. The spatial organization of the spiking sequences evolves to match the direction of pathological discharges, which spread over the cortex as traveling waves. Our data, therefore, suggest that seizures entrain cortical spiking sequences and change the spatial organization of firing across populations of neurons.

Clinical symptoms of seizures likely stem directly from the immediate impact of seizures on neural coding. There may be several mechanisms through which seizures disrupt normal brain function. For example, seizures can disrupt neuronal firing rates and give rise to synchronized, phase-locked bursts of spiking activity[3,20,26,27]. Hippocampal seizure activity may induce pathologic spindle oscillations in

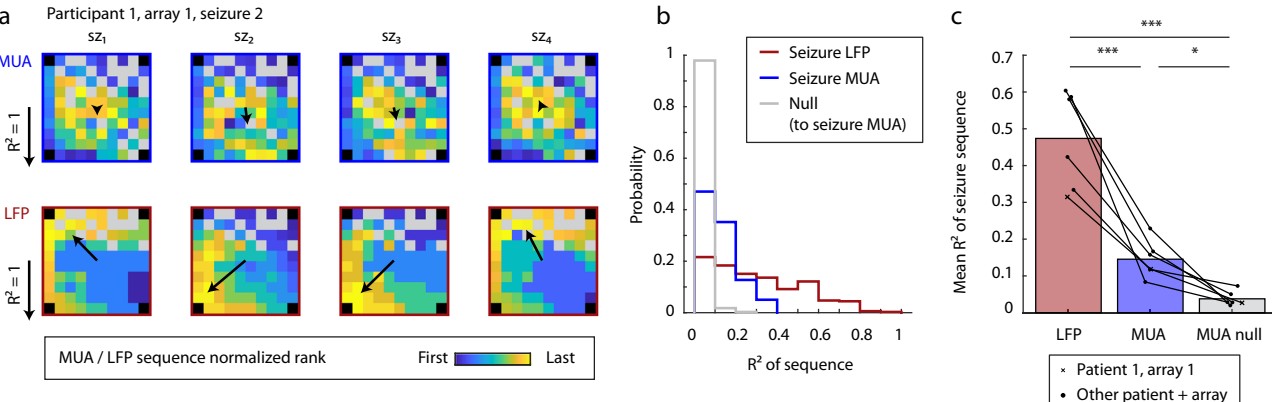

**Fig. 5 | LFP discharge sequences exhibit strong directionality. a** The timing of peak multi-unit spiking activity (MUA; top) and local field potential discharges (LFP; bottom) was mapped onto the spatial layout of the MEA during the same four example seizure bursts (blue: early; yellow: late). We fit a plane to the spatial organization of each spiking sequence, generating a measure of goodness of fit ($R^2$, indicated by the length of the black arrow) and a direction (indicated by the direction of the black arrow). **b** Distribution of $R^2$ values for all MUA spiking sequences (dark blue, mean $R^2 = 0.12 \pm 0.09$) and sequences of LFP discharges (maroon, mean $R^2 = 0.30 \pm 0.21$) during all seizure bursts in the same example seizure histogram, compared to null bursts (gray), created by scrambling the order of active spiking electrodes in the MUA sequences. **c** Average $R^2$ over all seizure bursts for LFP discharges (mean $0.47 \pm 0.13$), MUA spiking sequences (mean $0.15 \pm 0.05$), and null bursts in every participant and array. Each dot represents a single participant and array. Average $R^2$ was significantly greater for LFP discharges compared to MUA spiking sequences (two-tailed $p < 0.001$, repeated measures ANOVA; Holm–Bonferroni correction for multiple comparisons) and compared to null ($p < 0.001$). Average $R^2$ was also significantly greater for MUA spiking sequences compared to null ($p = 0.045$).

the prefrontal cortex[28]. Seizures may modify action potential waveforms, implicating changes in tissue micro-environments[29]. Seizures may also disrupt consciousness through the involvement of subcortical structures[30,31]. Any or all of these mechanisms may lead to clinical symptoms.

Our data provide an additional possible mechanism by which seizures may disrupt normal coding: the spatial entrainment of spiking sequences across neuronal populations. Sequence coding, in which the temporal order of spiking across a population of neurons encodes information, has emerged in recent years as a possible complement to strictly rate-based neural coding[15–19]. The emergence of stereotyped and directional sequences during seizures would severely constrain the coding capacity of such a sequence-based paradigm. In our data, we find such stereotyped sequences during all seizures in all participants. Because it is difficult to track the activity of single units during seizures[29], we instead analyze spatiotemporal sequences of neuronal activation by capturing the times of peak MUA at each electrode. Although we only track MUA, if sequences of MUA are disrupted, so too must be the underlying sequences of single-unit activity. Our data demonstrate that seizures profoundly constrain the flexibility of spike sequences, thereby providing an additional mechanism by which seizures may impact neural coding.

Our data also suggest that the entrainment of these spiking sequences may be shaped by the pathological discharges observed in the LFP. Recent evidence has demonstrated that pathological discharges, both during seizures and during interictal discharges, often propagate across the brain in the form of traveling waves[3–8]. Traveling waves are, in fact, commonly observed in the brain in the normal state, and the heterogeneity of wave direction may support cognition[32–35]. During pathological states, on the other hand, traveling waves are stereotyped and generally exhibit a consistent direction[7,20,25]. We thus examined whether the spike sequences we observed during seizures, particularly given their strong internal consistency, are related to these pathologic traveling waves. Because our recorded neurons occupy different spatial locations within a small patch of the cerebral cortex, sequences of spiking activity can also be characterized by their spatial organization.

We found that, as seizures unfold over time, the spatial organization of spiking sequences evolves to match the direction of the

pathological traveling waves in the LFP. The pathological discharges have a relatively strong and stable direction from the outset, related to the location of the MEA with respect to the source of the discharges (Supplementary Fig. 12)[5,7]. Spike sequences, on the other hand, resemble the sequences observed at baseline at seizure onset, at which time they do not exhibit a strong directional tendency. Abruptly after seizure onset, spike sequence consistency and stereotypy emerge (Figs. 2 and 3), along with spike sequence directionality (Fig. 4). Over the course of the seizure, often later in the event, spike sequence directionality evolves to match the direction of the traveling waves (Figs. 6 and 7). The time discrepancy between sequence stereotypy, on the one hand, and directional evolution, on the other, may simply reflect the fact that the latter is the end result of the former (Fig. 8). Our results, therefore, suggest that pathological discharges shape both the temporal order and the spatial organization of the spiking sequences. These results are consistent with findings in animal studies that suggest that excitation in hippocampal structures can entrain population spiking activity in surrounding regions[36].

Our results may shed light on a longstanding question in epilepsy: whether pathological activity observed in the LFP is merely a consequence of abnormal underlying spiking activity[24,37,38], or whether these discharges additionally induce local changes at the microscopic scale[39–41]. The direction of causality between LFP discharges and underlying spiking activity is difficult to assess, and similar uncertainty exists in the study of normal brain function[32–34]. Pathological discharges reflect post-synaptic potentials of tens of thousands of neurons, and may spread synaptically[36,41–43] or through non-synaptic mechanisms[36,41,43–46]. Once they have reached a remote region, pathological discharges may increase local neuronal excitability by providing subthreshold depolarization[32,39,40,47]. Moreover, prolonged and repeated exposure to these discharges may break down local inhibitory restraint[3,20,29,48,49]. Although we cannot resolve the question of causality with certainty, our data suggest that macroscopic activity influences microscopic networks. We find that after a brief period of exposure, spiking sequences are influenced by pathological discharges, changing their order and taking on directional features.

Together, our data demonstrate that spiking sequences during seizures are highly constrained. Moreover, as spiking sequences become more self-similar and diverge from baseline, they meanwhile

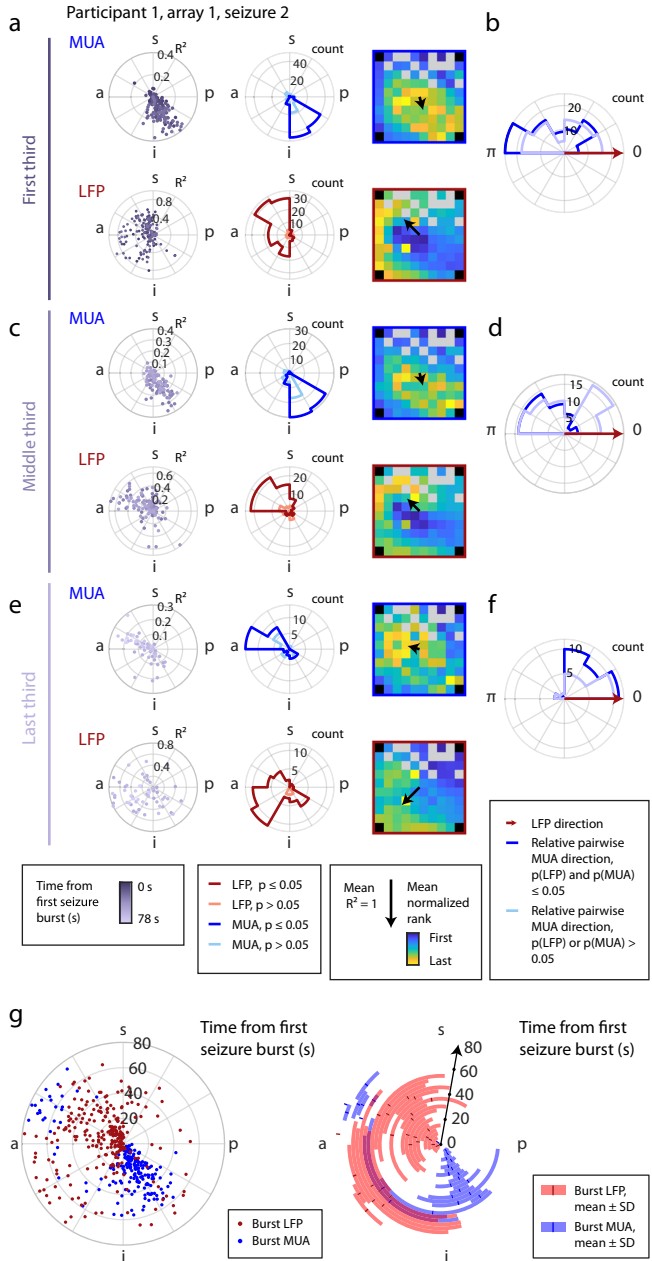

**Fig. 6 | Direction of spiking bursts evolves to reflect direction of LFP discharges over the course of the seizure. a** Left: direction and $R^2$ for best fit of spatial linear regression to multi-unit activity spiking sequences (MUA; top) and local field potential discharges (LFP; bottom) for every seizure burst in the first third of the same example seizure. Each dot represents a single seizure burst. Bursts are color-coded based on the time following first seizure burst (dark purple to light purple for early to late bursts). Middle: Distribution of directions generated from spatial fits of MUA spiking sequences (top) and LFP discharge sequences (bottom) during every seizure burst in the first third of the example seizure. Distribution of directions for bursts with significant fit to the plane (two-tailed $p \leq 0.05$) are shown in darker colors, while bursts that did not have significant fit are shown in lighter colors. Right: average temporal sequence of MUA (top) and LFP discharges (bottom) mapped to the spatial layout of the MEA. Mean $R^2$ of spatial fit is indicated with the black arrow. **b** Distribution of angular differences between the direction of the MUA spiking sequence and the direction of the sequence of LFP discharges across all seizure bursts in the first third of the seizure. Angular differences are shown relative to the angle of the LFP sequences (maroon arrow set to 0). **c–f** Same as above but for all seizure bursts in the middle and final thirds of this example seizure. In the final third of the seizure, MUA direction changes from inferior and posterior to anterior and superior, mirroring the LFP direction. **g** Left: for each burst in this example seizure, we plotted sequence direction ($\theta$ axis) against time from first seizure burst ($\rho$ axis). Only spike sequences (dark blue) and LFP discharge sequences (maroon) that had a significant fit with a spatial linear regression (two-tailed $p \leq 0.05$) were included. Right: for each three-second bin in this example seizure, we identified the mean (lines) and standard deviation (shaded error bar) of the direction of the spike sequences (dark blue) and LFP discharge sequences (maroon) that had a significant spatial fit.

## Methods
### Participants

We recruited five participants with drug-resistant epilepsy to participate in this study ($27.80 \pm 5.76$ years old, all male; see exclusions below, Supplementary Table 1). Participants underwent a surgical procedure for placement of intracranial electrodes with platinum electrode contacts (PMT Corporation, Chanhassen, MN, USA) for monitoring potential epileptogenic regions. Electrodes were implanted subdurally on the cortical surface and/or deep within the brain parenchyma (stereo EEG). Pre-surgical evaluations of each participant suggested a potential seizure onset zone in the temporal lobe in four participants and in the right parietal lobe in one participant. As such, in each participant, we placed an MEA (Cereplex SI; Blackrock Microsystems, Salt Lake City, UT, USA) in the regions suspected to be involved in seizure origin or spread in order to record multi-unit spiking activity during seizures. Each MEA contains individual micro-electrodes spaced 400 µm apart and extending 1 mm into the cortex. In three participants, we placed a single 96-channel MEA ($4 \times 4$ mm) in the middle temporal gyrus of the anterior temporal lobe, ~2–4 cm from the temporal pole. In one participant, we placed a single 96-channel MEA in the right parietal lobe (participant 5). In one participant, we placed two 64-channel MEAs ($3.2 \times 3.2$ mm) in the anterior temporal lobe ~1–2 cm apart from one another and also ~2–4 cm from the temporal pole (participant 2). Since one of the participants had two MEAs, our analysis includes six MEAs. We considered each MEA as an independent sample for our analyses since they recorded spiking activity from different populations of neurons. All patients were in the third or fourth decade of life (Supplementary Table 1). Since their ages were similar, we wouldn't expect age to act as a meaningful covariate. On the other hand, this may limit generalizability to patients older or younger than those included in our study.

In each case, we selected the implant site to fall within the expected resection area, but where no structural abnormalities were identified based on pre-operative MRI and visual inspection during the implant surgery. Intracranial electrodes may be placed so as to record from regions directly involved in seizure onset and spread, as well as from regions that may be unaffected by seizures—consistent with

become more directional, so as to match the direction of pathological traveling waves. Thus, our data suggest that traveling pathological discharges in focal epilepsy entrain sequences of spiking activity in local neural populations, thereby limiting the flexibility of neural coding. Whether such disruptions are directly related to the onset of clinical symptoms observed during seizures, however, remains unclear. Our recordings only capture the entrainment of spiking sequences in one small patch of cortex. In most cases, this is in the anterior temporal lobe, which may play a role in semantic cognition[50–52], but whose disruption may not be clinically salient. The entrainment process we observe here may, in fact, also transpire in other brain regions that underlie critical functions, the disruption of which may give rise to observed seizure symptoms. However, the timing relationship between entrainment in different brain regions cannot be assessed with our methods. Although our study cannot resolve the relationship between spiking entrainment and clinical symptoms definitively, the disruption of normal patterns of neural activity that we demonstrate here may nonetheless provide insight into a complementary mechanism by which seizures create symptoms.

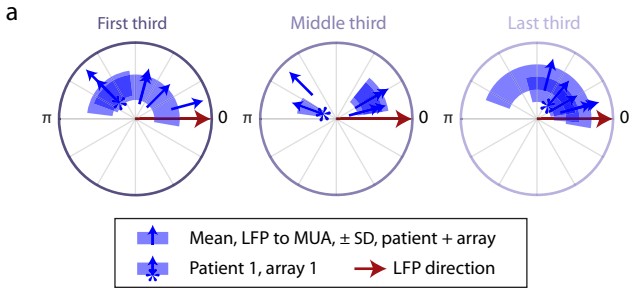

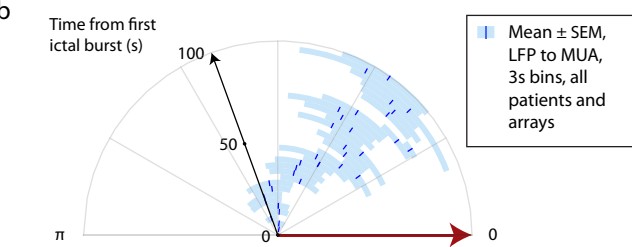

**Fig. 7 | Spiking burst direction conforms to LFP direction across seizures and patients. a** For each burst in each seizure, multi-unit activity (MUA) and local field potential (LFP) discharge directions were captured, and the difference between these two directions were retained for analysis. Each seizure for each patient was broken up into thirds. For each third, we obtained the mode of the distribution of MUA-to-LFP angular differences that had a significant spatial fit (for instance, see Fig. 6b, d, f, dark blue histograms). These modes were then averaged across all seizures separately in each participant. Here, each dark blue arrow reflects a single participant and array ($n = 6$ arrays; shaded error bar indicates standard deviation). Averaged mode directions are relative to the angle of the LFP sequences (maroon arrow set to 0). For each participant and array, distance from the origin (along the radial axis) is jittered to facilitate visualization. For the first and middle thirds, relative MUA directions are variable with respect to the LFP direction, but then coalesce towards the LFP direction in the final third. **b** Average angular difference between directions of all MUA and LFP discharge sequences with significant fits in every 3-second bin across all seizures in each participant. Mean (lines) and standard deviation (shaded error bar) across the six arrays are shown for each three-second bin, as a function of time following the first seizure discharge (radial axis). Over the course of seizures, mean MUA tends towards the direction of the sequence of LFP discharges (maroon arrow set to 0).

standard clinical practice. Therefore, the MEA may or may not record from tissue directly involved in seizures. Subsequent surgical resection boundaries are determined by anatomic and functional constraints, as well as by the clinical team's best estimate of seizure localization. Of the five participants initially recruited to the study, four received a surgical resection that included the tissue where the MEAs were implanted. The remaining participant received a surgical resection that involved regions posterior to the implanted MEA in the right parietal lobe. We did not note any changes in cognitive functions (e.g., vision, language, memory) or any unexpected changes in structural MRI imaging during post-surgical follow-up evaluations.

We performed all surgical procedures and iEEG monitoring at the Clinical Center at the National Institutes of Health (NIH; Bethesda, MD). All participants consented to participate in research, and consented to the procedures performed, including the placement of the MEAs under an NIH IRB-approved research protocol (reference number 11-N-0051).

### Micro-electrode recordings and multi-unit activity

We simultaneously recorded iEEG macro-electrode signals and MEA micro-electrode signals in all participants during the entirety of each participant's stay in the Epilepsy Monitoring Unit. All five participants retained for analysis had subdural electrodes arranged in both grid and

strip configurations with an inter-electrode spacing of 10 mm ($110.20 \pm 33.54$ electrode contacts per participant). Three of the five participants (participants 2, 3, and 4) also had depth electrodes with inter-electrode spacing of 3.5 mm ($28.67 \pm 18.58$ electrode contacts per participant). We sampled continuous macro-electrode iEEG data at 1000 Hz and re-referenced the raw signals in each participant using a global average reference. We used a regression-based approach to remove line noise at 60 Hz and 120 Hz[53].

The iEEG data acquisition system divides the continuous iEEG data into two-hour epochs. A clinical epileptologist experienced in the review of video-iEEG (S.K.I.) selected iEEG epochs for analysis. We selected two types of epochs: ictal epochs, which contained seizures, and interictal epochs, during which the participant was awake and resting quietly, or asleep, for the duration of the epoch, and during which no seizures occurred. The clinical epileptologist marked awake versus asleep states based on video-iEEG review. An effort was taken to ensure that, in each participant, a roughly equal number of awake versus asleep epochs were chosen. Interictal epochs were separated by at least 6 hours from a seizure event[5,7].

In total, we captured $3.60 \pm 2.70$ seizures per participant, with an average duration of $243.51 \pm 217.06$ seconds per seizure. In one participant (participant 3), we only captured one seizure. In the remaining participants, the range between the first and last seizure was, on average, $22.50 \pm 21.07$ hours. We also extracted for analysis $6.20 \pm 2.28$ interictal epochs per participant, comprising $329.40 \pm 152.23$ minutes of interictal recordings. The range between the first and last interictal epoch was, on average, $161.28 \pm 122.64$ hours. Of these, $2.60 \pm 1.14$ epochs per participant were awake recordings, comprising $115.88 \pm 59.22$ minutes. For purposes of analysis, we considered each ictal epoch separately, but we concatenated all interictal epochs in each participant.

We examined MEA recordings during the selected epochs. We aligned MEA recordings to macro-electrode iEEG recordings using a pulse-coded signal delivered simultaneously to the digital input ports of both recording systems. We recorded micro-electrode signals at 30 kHz using a Cerebus acquisition system (Blackrock Microsystems), with 16-bit precision and a range of $\pm 8$ mV, and referenced the micro-electrode signal to a reference wire laying adjacent to the array over the cortical surface.

Previous studies have shown that it is challenging to track the activity of single units during seizures, because distinct single-unit waveforms are lost during seizures[29]. Single-unit waveforms are also lost over the course of multi-day recordings. In light of this, we considered MUA, rather than single-unit activity, during seizures, consistent with previous studies[3,20,25]. Therefore, rather than examining sequences of individual neurons, we instead examined the sequence in which anatomic locations were activated.

To extract MUA, we first globally re-referenced each electrode's signal offline by subtracting the mean signal of all the electrodes in the MEA. We then used a second-order Butterworth filter to bandpass the signal between 0.3 and 3 kHz. We $z$ scored the 0.3–3 kHz filtered MEA signal in each micro-electrode across the entire epoch. We defined MUA as any event in the filtered signal with an absolute value of the $z$ score exceeding four standard deviations. We additionally enforced a 1-ms refractory period[20]. Although sorted single-unit spike waveforms are not reliable over timespans greater than a single epoch, our analyses reveal consistent multi-unit activity across epochs spanning multiple days (Fig. 1). For analysis of micro-electrode LFP signals, we did not re-reference the signal and instead only applied a low-pass filter, with frequencies below 500 Hz in the passband, to the raw recorded trace.

We manually inspected MUA rasters and micro-LFP traces for any obvious artifacts. We discarded data from 13 micro-electrode channels in participant 1, and 7 micro-electrode channels in participant 2 (array 2). We did not discard data from any other channels in any other participant

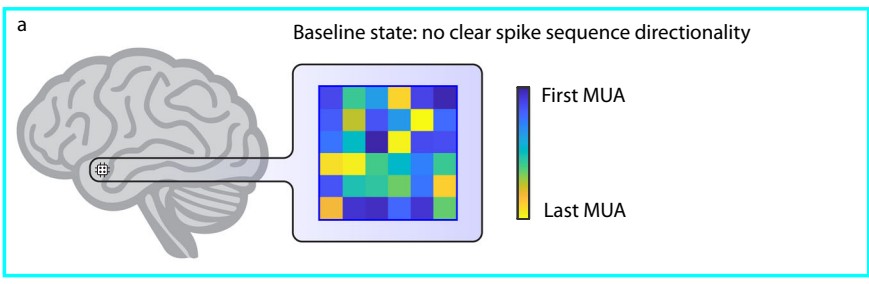

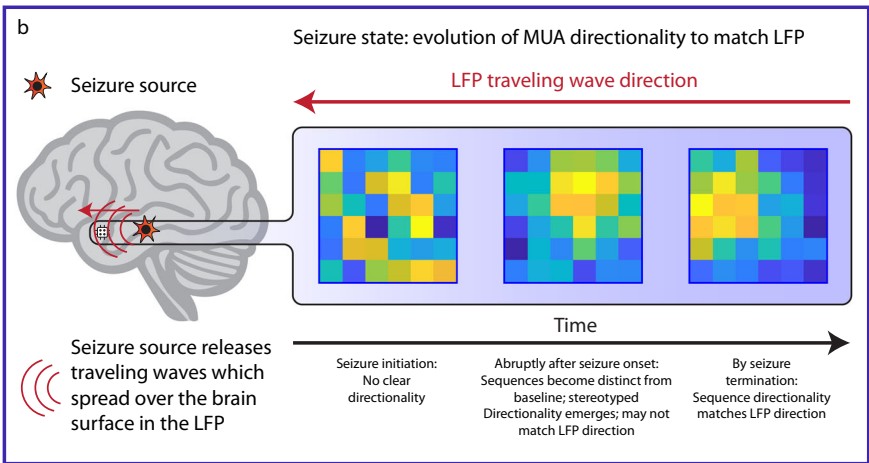

**Fig. 8 | Schematic: macroscopic ictal traveling waves in the LFP influence ordering and directionality of spiking sequences. a** In the baseline state, sequences generally do not have a clear directionality (Fig. 4c). **b** We conceptualize the seizure source as a focal entity, which releases discharges that spread outward over the brain surface as traveling waves (Fig. 5, Supplementary Fig. 12). As these discharges repeatedly spread to and over the array, they progressively alter underlying spike sequences. Very early in the seizure, MUA sequences generally exhibit weak or no directionality (inset, left array; see Fig. 4d). Abruptly after seizure onset, sequences become distinct from baseline (Fig. 2), stereotyped (Fig. 3), and take on directional properties (Fig. 4), although direction may not initially match the LFP direction (inset, middle array; see Fig. 6a–d). By the end of the seizure, MUA sequence direction tends to match LFP sequence direction (inset, right array; see Fig. 7).

or array. In addition, we also manually removed time periods in the recorded epochs that showed frequent and severe artifacts. These sections often lasted only several seconds, were never longer than several minutes, and did not occur during seizure events.

## MEA recruitment to seizures

Our analysis, which examines changes in MUA during seizures, requires that seizures involve the tissues underlying the MEA. We, therefore, only included seizures for subsequent analysis that demonstrated clear recruitment of the MEA[3,25]. We used three metrics to assess seizure recruitment.

First, in order to be considered recruited by a seizure, we required that the MEA demonstrate bursts of spiking activity during ictal discharges (see "Multi-unit activity burst detection" for criteria for selecting discharge-related bursts). We considered MEAs that did not demonstrate this phenomenon to not be recruited by the seizure.

Second, we used a previously described metric known as the weighted phase-lag index (WPLI) to assess seizure recruitment. This metric of phase coherence estimates the extent to which the signals in every pair of micro-electrodes demonstrate a consistent phase relation that is non-zero and not centered around $\pi$, thus reducing the likelihood that such phase relations are due to volume conduction. WPLI is computed from the imaginary component of the power spectrum. During seizures, the highly synchronized activity across electrodes yields an elevated WPLI[3]. We used an implementation from Fieldtrip (http://fieldtrip.fcdonders.nl/), using previously described parameters[3]. We assessed WPLI between every pair of micro-electrodes over two time periods: an early time period beginning 15 seconds prior to the electrographic onset of seizures as identified by the clinical epileptologist, and a late time period beginning 15 seconds after the first seizure burst. We computed the WPLI at six frequencies between 2 and 30 Hz (2, 7, 12, 17, 22, and 27 Hz) for each micro-electrode pair in the MEA. For each time period, we computed the WPLI between the time series of every micro-electrode pair over 1-second sliding windows stepped in 8-ms increments for a total of 20 steps. For every electrode pair, we computed the average WPLI across all time steps and across all frequencies separately for the two time periods. We then compared the distribution of WPLI values across all micro-electrode pairs between the two time periods. We considered an MEA to be recruited to seizures if the WPLI was significantly different between the two time periods (two-sample $t$ test, $p < 0.05$).

Third, we assessed seizure recruitment by examining spike-LFP correlations. For each seizure, we captured the timestamps of all discharge-related bursts (see "Multi-unit activity burst detection"). We filtered the ictal LFP between 2 and 12 Hz, and applied the Hilbert transform to the filtered data. We extracted phase and magnitude information, and then captured the mean phase of the signal at the time of each burst. In a recruited seizure, the burst-related phases should be similar across bursts. To test this, we used the Rayleigh test for non-uniformity of circular data, implemented in the circular statistics toolbox (http://www.jstatsoft.org/v31/i10)[54]. We considered an MEA to be recruited to seizures if the distribution of phases was significantly non-uniform ($p < 0.05$).

We considered an MEA to be recruited to seizures only if all three metrics above were satisfied. For an example of WPLI and spike-LFP correlation results that demonstrate recruitment, see Supplementary Fig. 24. This is the same exemplary participant, array, and seizure considered in the figures of the main analysis. Similar figures

demonstrating recruitment are provided for all seizures included in this study in the Supplementary Figures. We considered one additional participant for inclusion in our study, but in this individual, no seizures were recruited to the MEA. We, therefore, did not include this participant in the data that we present here. In addition, in some participants, only a subset of seizures exhibited recruitment to the MEA (Supplementary Table 2). All recruited seizures were either focal impaired awareness seizures or focal to bilateral generalized tonic-clonic seizures. Details of seizure count, type, and recruitment status are provided in Supplementary Table 2. We focused our analyses on these recruited seizures, which, in total, comprised $2.50 \pm 1.76$ recruited seizures per array with an average duration of $230.25 \pm 195.05$ seconds per seizure.

## Ictal and interictal discharge detection

We used a previously published automated procedure to detect ictal and IEDs in the macro- and micro-electrode contacts[7,55]. Briefly, we filtered the broadband iEEG or LFP signal between 2 and 50 Hz to incorporate low-frequency, high-amplitude activity characteristic of ictal discharges, while excluding high-frequency activity that may reflect neuronal spiking activity[3,5,56–59]. We identified all instances in which there was both an upward and downward deflection of at least $3\sigma$ within the same 150 ms window. We also required a peak-trough difference of $\geq 9\sigma$. We marked the time of the downward peak of each of these events. In order to mark an IED on the MEA for further analysis, we required that IEDs be present on at least ten individual micro-electrodes within 30 ms. In that case, we assigned the timestamp of the discharge event as the median timestamp of the individual IEDs.

## Multi-unit activity burst detection

To detect bursts of spiking activity, we constructed spike rasters based on the times of the captured MUA for each two-hour epoch. We downsampled the spike trains from the original 30 kHz sampling rate to 2 kHz. We smoothed each micro-electrode channel's spike train using a Gaussian kernel of width 25 ms[17], and then computed the mean smoothed firing rate across all micro-electrodes (Fig. 1a). For each participant, we concatenated all interictal two-hour epochs to generate a time series of mean smoothed spike rate across all interictal epochs. We identified the spiking rate corresponding to five standard deviations above the mean of this time series. We used this absolute spiking rate as a threshold for detecting spiking bursts during all interictal epochs, as well as in all ictal epochs, for that particular participant. We marked every time the mean smoothed spiking rate exceeded this threshold as a spiking burst event.

In interictal epochs, we used a burst width of 150 ms, and required that interictal burst centers be $\geq 150$ ms from each other. In ictal epochs, we used a minimum inter-burst interval of 75 ms, since bursts occurred in short succession in these epochs, and a narrower burst width of 100 ms, to reduce the likelihood that adjacent bursts overlap. The rate of high-amplitude, low-frequency seizure discharges typically does not exceed 12 Hz, corresponding to an inter-discharge interval of 83 ms[5]. Therefore, an inter-burst interval of 75 ms would still be short enough to allow for the capture of bursts associated with a 12 Hz ictal rhythm.

In some participants, we noted the presence of artifactual bursts that appeared to involve simultaneous MUA in almost every micro-electrode (Supplementary Fig. 1c). The source of this artifact is unclear. We removed these artifacts using an automated approach. For each detected burst, we computed the cross-correlation between all electrode pairs (excluding autocorrelations) over the timespan of each burst and then computed the average cross-correlation over all pairs. We fit this average cross-correlation with a Gaussian. We obtained the goodness of fit, $R^2$, as well as the standard deviation of the best fit. We rejected each detected burst if the $R^2$ was less than 0.75 or if the standard deviation was less than 10 samples (5 ms). We selected these criteria because physiological bursts typically fit well with a Gaussian distribution. On the other hand, the observed artifactual events usually fit poorly with a simple Gaussian distribution. In some cases, these artifacts contained a single instance in which there was nearly simultaneous MUA on almost all electrodes. In those cases, while fit to the Gaussian may be good, the Gaussian curve was very narrow. Based on these criteria, we rejected a variable proportion of bursts as artifactual in each participant, suggesting that artifact rejection was not simply removing a fixed proportion of bursts in all participants (Supplementary Table 3).

We linked bursts of spiking activity during interictal epochs with interictal discharges that occurred within 150 ms of the identified spike burst. We also linked spiking bursts during ictal epochs with ictal discharges that occurred within 75 ms of the identified spike burst. In rare cases, a single spike burst may be linked with two ictal discharge events. This can be possible, for example, when the discharge waveform is complex (see burst $sz_3$ in Fig. 1a as an example). In these cases, we linked the spike burst with the discharge that occurred closest in time to the center of the spike burst. Based on this mapping, we, therefore, identified two types of spiking bursts during interictal epochs: spiking bursts with no accompanying IED, which we refer to as baseline bursts, and spiking bursts that are linked with interictal discharges, which we refer to as IED bursts (Supplementary Table 3). In our main analysis, for ictal epochs, we only considered discharge-related bursts, which we refer to as seizure bursts. In Supplementary Fig. 5, we additionally considered interictal bursts and baseline bursts which occurred in ictal epochs. These bursts were classified as pre-ictal IEDs or pre-ictal baselines, if they occurred before the seizure, and post-ictal IEDs or post-ictal baselines, if they occurred after the seizure.

We were interested in determining false discovery rates for our burst detection protocol. To this end, we again produced MUA rasters for each interictal epoch in our study. We then constructed surrogate data by scrambling the latencies between multi-unit action potentials for each raster and for each micro-electrode. In this way, we retained total spike counts for each micro-electrode and for the population, while scrambling the temporal organization of multi-unit action potentials[60]. Bursts were then detected as before, but on the scrambled rasters. False discovery rate of interictal bursts was zero, for all participants and arrays.

## Burst sequence similarity

We were interested in determining the similarity of sequences of MUA across different discharges. To do so, for each micro-electrode during each spiking burst, we determined the time of peak height of smoothed MUA (using a Gaussian kernel, 25 ms width). We then captured the times corresponding to the peak spiking activity in each micro-electrode, across the MEA, for that burst. We can represent the timings of spiking activity across the MEA in each burst as a vector $\mathbf{t} := (t_1, \ldots, t_n) \in \mathbb{R}_+^n$ of length $n$ micro-electrodes. We assigned all micro-electrodes that did not exhibit MUA during a burst a value of NaN (not a number) in that corresponding vector. We then compared the ordering of MUA timestamps, during any two bursts, using Spearman's rank correlation. This allowed us to measure sequence similarity between any two spiking bursts. In many comparisons between two bursts, some values for MUA activity were missing (NaN) in at least one of the bursts. We, therefore, removed the indices corresponding to those micro-electrodes from both vectors for that comparison. Therefore, when comparing two bursts from $n$ micro-electrodes, we created two vectors of MUA timestamps, each of length $m$, where $m \leq n$, and where both vectors contained no NaN values. For every comparison included in our analyses, we required that $m \geq 5$. Otherwise, comparisons were discarded. For every comparison between pairs of spiking bursts, we therefore generated a correlation coefficient $\rho$. We applied the Fisher $z$-transformation to the resulting correlation coefficient, which we then used for subsequent analyses.

To create a random null version of each observed sequence, for each burst, we shuffled the identified timestamps across the micro-electrodes in which MUA was detected. Shuffling the timestamps of only the active electrodes in each burst thus retained the number and indices of missing values.

## Dimensionality reduction

Dimensionality reduction has emerged in recent years as a powerful method for revealing patterns in complex, high-dimensional neural datasets[22,23]. Here, we used principal component analysis (PCA) for dimensionality reduction in order to examine patterns among the sequences of spiking activity across micro-electrodes.

To reduce dimensionality and, therefore, visualize spiking sequences in a lower dimensional space, we had to address specific challenges. First, virtually every burst of spiking activity includes missing timestamp data because not every micro-electrode participates in every burst. Second, it is challenging to represent rank data in a manner that is amenable to dimensionality reduction. The naive approach of simply inputting raw MUA timestamps, or normalized ranks, may confuse the dimensionality reduction procedure into treating the data as continuous or ordinal data. In reality, we were more concerned about relative differences in rank.

We addressed these concerns using the following approach. We performed dimensionality reduction separately for each seizure. For any one particular seizure, we collected seizure bursts in addition to all baseline and IED bursts for that participant. Therefore, for every seizure, we collected a set of $k$ bursts, where $k$ is the number of combined baseline, IED, and seizure bursts. Each burst contains a set of $n$ timestamps, corresponding to $n$ micro-electrodes, where each timestamp indicates the timing of that micro-electrode's peak MUA. Instead of using raw MUA timestamps, we used the normalized rank of peak MUA timing for each electrode within that burst, for normalized rank values $\in [0,1]$. Because dimensionality reduction procedures do not work with missing values, we in-filled missing normalized rank values using numbers from the standard uniform random distribution on [0,1]. This ensures that the indices of missing timestamps and the value assigned to them did not systematically influence the results.

We then created a matrix in *rank space* for each seizure in the following manner. First, we enumerated, in lexicographic order, the set of all 2-element subsets of micro-electrodes $\{1, \ldots, n\}$, yielding the set of tuples $\mathbf{p} := \{(1, 2), (1, 3), \ldots, (2, 3), (2, 4), \ldots, (n-1, n)\}$, of length $\binom{n}{2}$. We then created a sequence matrix with $\binom{n}{2}$ columns, corresponding to the number of tuples in the set $\mathbf{p}$, and $k$ rows, corresponding to the number of combined bursts:

$$A := \begin{bmatrix} \alpha_{1,(1,2)} & \cdots & \alpha_{1,(n-1,n)} \\ \vdots & \ddots & \vdots \\ \alpha_{k,(1,2)} & \cdots & \alpha_{k,(n-1,n)} \end{bmatrix} \quad (1)$$

Each column of this matrix, therefore, reflects a distinct pair of electrodes rather than a single electrode. We populated each value in each row with either a 0 or a 1, depending on whether the first member of the micro-electrode pair outranked the second in that particular burst. Hence, for each burst $i$ containing a set of timestamps, $\mathbf{t}_i = (t_{i,1}, \ldots, t_{i,n})$ and each pair of distinct electrode indices $(j_0, j_1) \in \mathbf{p}$:

$$\alpha_{i,(j_0,j_1)} := \begin{cases} 1 & \text{if } \mathbf{t}_{i,j_0} \leq \mathbf{t}_{i,j_1} \\ 0 & \text{if } \mathbf{t}_{i,j_0} > \mathbf{t}_{i,j_1} \end{cases} \quad (2)$$

This approach forces the dimensionality reduction procedure to consider relative rank information, rather than raw ordinal or continuous data. When considering null versions of sequences, we used the same procedure applied to a random version of a particular set of bursts. In these random sequences, we shuffled MUA timestamps at active electrodes were shuffled. We again in-filled missing values for normalized ranks with random values from the uniform random distribution on [0,1], prior to creation of $A$.

## Directionality of spike bursts and LFP discharges

We were interested in capturing the extent to which bursts of MUA and LFP discharges across the MEA were directional. For directional analyses of MUA, we extracted the timestamps of peak MUA activity across microelectrodes. For directional analyses of LFP discharges, we extracted the time of the negative peak of the discharges across microelectrodes. We then used linear regression to fit a plane to the spatial organization of the temporal pattern of timestamps. Effectively, the timestamps serve as heights, distributed across spatial coordinates, and we fit a plane to these heights. Similar approaches for spatial linear regression have been used in previous work for traveling wave analysis in epilepsy[25,61].

We extracted the velocity and direction of the spatial organization using the pseudoinverse of the regression coefficients[61]. We also retained the $R^2$ statistic, which captures how well the plane fits the empirical data. $R^2$ is a measure of the spatial directionality of the sequence. If timestamps are spatially organized so as to be monotonically increasing from one end of the MEA to the other, then a plane would fit the data perfectly, yielding $R^2 = 1$. On the other hand, if the data cannot be fit with a plane, then $R^2 = 0$. We also obtained the $p$ value of the regression, which represents the likelihood of observing $R^2$ equal to or greater than the empiric $R^2$ by chance alone.

## Statistical analyses

All data are reported as mean ± SD unless otherwise reported. We analyzed data using custom MATLAB scripts (Mathworks, Natick, MA, version R2023b). For $t$ tests, we used Cohen's $d$ to measure effect size. We computed statistics for repeated measures ANOVA using JASP[62]. For repeated measures ANOVA, we used partial $\eta^2$, again computed in JASP, to measure effect size. In all repeated measures ANOVA studies, we used the Holm-Bonferroni method for $p$ value correction in post-hoc comparisons. All $p$ values are two-tailed. Repeated measures ANOVA assumes normality and sphericity of underlying data. We used Mauchly's $W$ test to evaluate for sphericity prior to running repeated measures ANOVA. In instances where the sphericity assumption was violated, we used the Greenhouse-Geisser method to correct the $p$ value. For our analyses, we considered the two arrays that were implanted in one participant as two independent samples. Therefore six total arrays were included in the study. For summary statistics regarding seizure bursts, analyses were done first across bursts within each particular seizure, and the results were averaged across seizures to create an array-level statistic. Our results in Fig. 1d, e suggest that sequences are similar from seizure to seizure, thereby justifying this approach. For statistics involving baseline and IED bursts, we performed analyses on the full set of baseline or IED bursts for each participant. Therefore, we computed baseline, IED, and seizure results at the level of each MEA. We then performed descriptive statistics on the set of six data points corresponding to the six MEAs, across conditions. Each array may be thought of as a biological replicate and technical replicates were not considered in this study.

The Fisher $z$-transformation was applied to all computations of Spearman's $\rho$. This ensured that resulting $\rho$ values were approximately normally distributed. To aggregate correlations across seizures, we first computed Spearman's $\rho$ (with the Fisher z-transformation) from every correlation of interest (e.g., between the directionality, $R^2$, of each spiking burst and the distance in PCA space between those bursts and the centroid of the baseline bursts), for every seizure in every

array. We then weighted each correlation by the number of bursts in that seizure, to compute the weighted average correlation across seizures:

$$Z(\rho)_w = \frac{\sum_{i=1}^{k}\left((n_i - 3) \cdot Z(\rho)_i\right)}{\sum_{i=1}^{k}(n_i - 3)}. \qquad (3)$$

where $k$ is the number of seizures, $n$ is the number of bursts in each seizure, and $Z(\rho)$ is the $z$-transformed correlation for each seizure[63]. The resulting value is a summary statistic at the level of the array. We also pooled the variance of the correlations to obtain the aggregated variance of the correlation estimate:

$$V_w = \frac{\sum_{i=1}^{k}\left((n_i - 3) \cdot \left(\sqrt{\frac{1}{n_i-3}}\right)^2\right)}{\sum_{i=1}^{k}(n_i - 3)} = \frac{k}{\sum_{i=1}^{k}(n_i - 3)}. \qquad (4)$$

In this manner, we generated aggregated values of $Z(\rho)$ and $V$ across seizures in each array. For each array, the 95% confidence interval was then $Z(\rho)_w \pm 1.96 \cdot \sqrt{V_w}$.

### Reporting summary
Further information on research design is available in the Nature Portfolio Reporting Summary linked to this article.

## Data availability
Processed data used for analysis are publicly available for download at https://research.ninds.nih.gov/zaghloul-lab/downloads. Processed data are also provided with the manuscript. Finally, processed data are provided in a Code Ocean repository, available at https://doi.org/10.24433/CO.7738428.v2. Because of size constraints, raw data are not provided but are available upon request. Source data are provided with this paper.

## Code availability
All code used for analysis is publicly available for download at https://research.ninds.nih.gov/zaghloul-lab/downloads. Code is also provided with the manuscript. Finally, code is provided in a Code Ocean repository, available at https://doi.org/10.24433/CO.7738428.v2.

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

## Acknowledgements

This work was supported by the Intramural Research Program of the National Institute for Neurological Disorders and Stroke. The authors report no disclosures relevant to the manuscript. We thank Elliot Smith for sharing ideas with us. Alex Vaz provided ideas that motivated the initial explorations of the study. We are indebted to all patients who have selflessly volunteered their time to participate in this study.

## Author contributions

J.M.D., J.I.C., and K.A.Z. conceptualized the study. J.M.D., W.X., and K.A.Z. helped to develop the methodology. J.M.D. and S.N.J. wrote software. S.K.I., S.N.J., and K.A.Z. collected the data. J.M.D. and K.A.Z. analyzed the data and wrote the original draft. J.I.C., W.X., S.N.J., S.K.I., and K.A.Z. reviewed and edited the draft. K.A.Z. supervised the work.

## Funding

## Competing interests

The authors declare no competing interests.
