## [Peer Review File · Nature Communications]

Focal seizures induce spatiotemporally organized spiking activity in the human cortexREVIEWERS' COMMENTS

Reviewer #1 (Remarks to the Author):

This manuscript describes the organization of multi-unit activity in the neocortex of epilepsy patients during three separate conditions: seizures, interictal epileptiform discharges (IEDs) and population bursts not related to an LFP phenomenon. The authors analyze separately epochs with detected seizures and epochs that contain only interictal discharges (IEDs) or population bursts and demonstrate that along cortical multi electrode array (MEA), the spatial pattern of population firing in seizures is more stereotypical than in IEDs, whereas baseline population bursts show the highest diversity. An expected finding is that IEDs seem to be a heterogeneous group, similarly to baseline bursts. The authors further demonstrate that during seizure bursts, the firing rate patterns become more dissimilar from baseline bursts and more similar to each other. They also demonstrate that the propagation of the seizure LFP is initially not matched by the spatial pattern of neuronal firing rate, whereas MUA firing becomes more directional and similar to LFP direction in late seizure bursts. The analysis performed in the manuscript clearly distinguishes neuronal recruitment during IEDs and seizures, and demonstrates that the seizure duration seems relevant for the entrainment of neuronal firing.

The authors suggest that the main significance of the manuscript is to demonstrate the entrainment of neuronal firing sequences by seizures. However, there are several concerns:

The authors have chosen to refer to the order of MEA channels with peak MUA firing rate as firing sequences. This could be misleading as firing sequences is a term commonly used for sequential activation of single neurons in the neuroscience literature (with the exception of tissue culture physiology). In this case, however, the analysis focuses on the order of anatomical locations where neurons will be activated.

Spike bursts and the evolution of stereotypical population patterns have been described in the rat (Bragin et al. Neuroscience, 1997). It would be constructive if the authors attempted to compare their findings to animal work.

The authors are addressing whether ictal bursts lead to pathological entrainment. If this is the case, it should be reflected in post-seizure baseline or interictal bursts. However, in this study, interictal epochs were separated by at least 6 hours from ictal epochs (line 390). This separation may create an artificial difference of the brain state in which IEDs and normal bursts were analyzed. It would be useful to also analyze interictal bursts and baseline bursts preceding and following seizures and demonstrate how the postictal bursts are affected.

Furthermore the authors should differentiate between recording from seizure-onset-zone (SOZ) and functional cortex and report the respective recording location - this can only be inferred for the parietal participant, in whom apparently the recording site was outside the SOZ as it was not resected. If the cortex is already dysfunctional, it is difficult to argue that seizures will contribute to functional impairment by pathological entrainment.

In the discussion (line 338) the authors propose that the entrainment of spiking sequences may serve as a biomarker for seizure onset - however they do not show any data about the seizure onset zone - see also previous point. Moreover the authors suggest that returning to baseline sequences can be a biomarker for seizure termination - they should explain better what would be the benefit of such biomarkers.

For an audience outside the technical field it is very difficult to interpret the paper, especially figures 3 and 4. The reader needs to constantly switch between the main text and the methods section. As the findings of the paper may be interesting for non-technical readers, including clinical audience, the authors should find a way to better explain the analysis methods in the results, for example by adding cartoons to the figures, more detailed graphs labels and brief methods explanations, rather than solely referring to the methods section. The legends of the figures are very concise and often incomprehensible on their own without referring to the methods section. Also, see some examples in the minor comments, but the figure legends and labels should be generally much more informative.

minor points:

line 107 - the reference to figure S2 is wrong

fig 1a- the scale bar annotation (σ) is not clear for a non-technical reader and should be made more clear how σ is related to the mean firing rate (blue traces). Also what is the scale for the red traces? (LFP).

fig 1a - please comment on why the firing rate is highest for interictal bursts. Amplitude and time calibrations are missing.

Are fig. 1a and fig 3a identical traces?

lines 357 - 358 methods - the patient description is confusing as one of the listed participants was not included in the analysis. This participant seems irrelevant to the study and could be excluded. In summary three patients had a MEA implanted in medial temporal gyrus, one in temporal cortex but not in medial temporal gyrus (2 sessions) and one in parietal cortex. It is questionable whether the parietal recording should be pooled with the temporal recordings, especially since this patient did not have burst-related IDs (Table S3)

Table S1 - several abbreviations mentioned in the legend (LGG, FCD) - do not appear in the actual table. Also some abbreviations seem to be spelled out differently in the table and legend.

fig 2f is missing x and y labels

fig s1a - the legend seems unrelated to the figure and it is not clear what the blue and red bars in the right plot are

figs2 b/d - y axis - different scales or wrong numbers? the figure legend refers to r2 but the y-axis label is firing rate correlation, which is confusing

line 438 - the authors probably mean volume conduction (instead of condition)

please explain the sleep detection algorithm as the sleep duration is unusually high for epilepsy patients (line 396 : from 6.20 ± 2.28 interictal epochs per patient on average 2.60 ± 1.14 epochs per participant were awake)

Reviewer #2 (Remarks to the Author):

In this study, the authors recorded intracranial EEG from depth electrodes alongside single unit activity and 'micro-scale' LFP signals from six multi-electrode arrays (MEAs) in five patients with medically refractory focal epilepsy. They recorded between 1 and 6 seizures in each patient that 'recruited' single unit activity recorded on the implanted MEA, and then compared the spiking

dynamics of population burst events between these seizures, interictal epileptiform discharges (IEDs) and ‘baseline’ periods.

First, they find evidence for a consistent temporal structure in multi-unit activity across MEA contacts both within and between seizures – as indicated by a greater Spearman’s rank correlation compared to IED and baseline bursts. Next, they use the Uniform Manifold Approximation and Projection (UMAP) method to embed each spiking sequence in a low-dimensional space, further characterise the relationship between spiking sequences in different bursts, and track their evolution over the course of each seizure. They find that seizure burst sequences rapidly diverge from baseline sequences after seizure onset and become more consistent over time as they do so. Next, they show that seizure bursts tend to exhibit greater spatial structure than random null sequences, and that this spatial structure (or ‘directionality’) also increases over time along with distance from the centroid of baseline sequences in UMAP space. Finally, they show that the directionality of spiking activity only matches the directionality of the LFP signal in the final third of seizures.

Overall, the manuscript is clearly presented, the data presented are rare and valuable, and the analyses appear to be sound. However, the authors make little or no attempt to link their neural data to behaviour, and it is difficult to know what theoretical contribution is made by this study – i.e. what we can learn about epilepsy, neural coding, or cognitive function from these results. These concerns are described in more detail below.

Main Concerns

I agree that temporal coding is an interesting and often overlooked phenomena in single unit recordings across species. However, from the examples shown in Figure 1, it also seems that there are clear firing rate differences between multi-unit bursts of activity occurring during different time periods of interest (seizures, IEDs, and baseline) in these data, but these differences are overlooked in the analyses and results. From a methodological perspective, it would be useful to demonstrate that differences in firing rate do not introduce a confound into the temporal sequence analyses (i.e. by down-sampling the spiking data so that firing rates are equal across bursts from different time periods of interest and then repeating the burst sequence analyses). From a theoretical perspective, it would also be useful to characterise these differences in firing rate more clearly by making formal comparisons between mean firing rates during bursts in each period, and by analysing the consistency of firing rates between bursts in the same and different periods.

Figure 2 could be much clearer. First, I am not sure that varying the colour code used for seizure bursts to indicate the time since the first seizure burst in panel A makes these results particularly easy to interpret. It might be preferable to first show the three different types of burst (seizure, IED,

baseline) in three strongly contrasting colours, and then show the time evolution of distance from the baseline centroid in panel C only. Or perhaps, show a second version of this plot with the three second moving average of an example seizure trajectory marked in UMAP space as a thick black line with an arrowhead? Second, it might be useful to add the mean distance from baseline bursts to the baseline centroid from Fig 2b as a red dashed line onto Figure 2c, to illustrate the evolution of seizure bursts in relation to the baseline centroid more clearly. Finally, it might be worth using a separate, distinctive colour map for the Spearman's rho values in Fig 2f, to dissociate those from the colour code indicating time since the first seizure burst in all other panels

There is little mechanistic interpretation or attempt to link the neural data to behaviour or clinical symptoms. In the Discussion, the authors “suggest that ... constraints on the flexibility of spike sequences may contribute to disruptions in normal cognitive processes”, but in the caption of Figure S10, they state that “clinical symptoms begin before or within seconds after the first seizure-related burst” – i.e. before burst sequences become disrupted and the flexible temporal code observed during baseline periods is constrained. These two statements seem to be completely contradictory. What we are left with is a fairly complex story about the time evolving impact of seizure activity on neural firing, with no real idea of what this means either for understanding the cognitive impact of epilepsy, seizure generation, or ‘healthy’ brain function. As such, it is difficult to identify the implication of these results, or what conclusions should be drawn from these findings.

Minor Comments

Lines 116-118: The following statement could be phrased more clearly: “Spike sequence similarity among seizure bursts was significantly greater than similarity between seizure bursts, IED bursts, and baseline bursts”, as it seems to imply that “similarity among seizure bursts was ... greater than ... similarity between seizure bursts”, which is confusing

Lines 132-136: To confirm that burst sequences during seizures occupy a different part of the low-dimensional subspace than those that occur during baseline, the authors compare the UMAP distance between baseline bursts and the baseline centroid with the UMAP distance between seizure bursts and the baseline centroid. Could they also include IED burst sequences in this comparison, or provide a compelling reason why not?

Line 145: Around this point, it would be useful to provide details regarding the duration of seizures across patients, to provide some context for the use of a 3s moving average

Lines 148/149: Admittedly, this is a facetious point, but there is no necessary reason why the maximum (minimum) distance would be greater (less) than the initial and final distances – for example, if the distance from the baseline increased or decreased monotonically over time

Lines 197-200: Can the authors test whether seizure bursts exhibited greater spatial structure (i.e. were better fit by a plane) than bursts that occur during IEDs or baseline periods? I think this would be particularly relevant to the interpretation of these findings

Lines 278/279: “As the seizure evolves, spiking sequences diverge from the sequences observed ... during interictal discharges” – I am not sure that this result is actually shown anywhere in the manuscript, the authors only show that spiking sequences diverge from baseline. Please edit accordingly

Methods, multi-unit activity burst detection: It seems concerning that different criteria are used to identify bursts during different periods, which are then compared as like for like in the main manuscript. Can the authors demonstrate that the main results are qualitatively unaffected by using consistent burst detection criteria?

REVIEWER COMMENTS

Reviewer #1 (Remarks to the Author):

This manuscript describes the organization of multi-unit activity in the neocortex of epilepsy patients during three separate conditions: seizures, interictal epileptiform discharges (IEDs) and population bursts not related to an LFP phenomenon. The authors analyze separately epochs with detected seizures and epochs that contain only interictal discharges (IEDs) or population bursts and demonstrate that along cortical multi electrode array (MEA), the spatial pattern of population firing in seizures is more stereotypical than in IEDs, whereas baseline population bursts show the highest diversity. An expected finding is that IEDs seem to be a heterogeneous group, similarly to baseline bursts. The authors further demonstrate that during seizure bursts, the firing rate patterns become more dissimilar from baseline bursts and more similar to each other. They also demonstrate that the propagation of the seizure LFP is initially not matched by the spatial pattern of neuronal firing rate, whereas MUA firing becomes more directional and similar to LFP direction in late seizure bursts. The analysis performed in the manuscript clearly distinguishes neuronal recruitment during IEDs and seizures, and demonstrates that the seizure duration seems relevant for the entrainment of neuronal firing. The authors suggest that the main significance of the manuscript is to demonstrate the entrainment of neuronal firing sequences by seizures. However, there are several concerns:

We thank the Reviewer for the careful review of our manuscript and for the constructive suggestions.

The authors have chosen to refer to the order of MEA channels with peak MUA firing rate as firing sequences. This could be misleading as firing sequences is a term commonly used for sequential activation of single neurons in the neuroscience literature (with the exception of tissue culture physiology). In this case, however, the analysis focuses on the order of anatomical locations where neurons will be activated.

We agree with the Reviewer that this could be misleading and acknowledge that this is a limitation of our approach. However, it would be difficult to surmount this limitation, given previous data that it is challenging or impossible to track single units during seizures. Therefore, we feel that these “sequences of anatomic activation” are the best possible surrogate we can get to canonical sequence data. We do note that, in traditional spike sorting, many electrodes only have one neuron. Therefore, it is likely that, at least in some cases, the mapping between single units and multi-units is one-to-one. Even in cases with multiple neurons per electrode, multi-unit sequences should serve as an approximation of single unit sequences. Particularly in seizure bursts, MUA timings tend to be fairly temporally-constrained (see Figure 1a), which suggests that using the peak MUA time as the single MUA time for that electrode can still provide reliable data.

We also note that we do find robust sequences of multi-unit activity even in the baseline state (Figure S4). The sequences are drastically-different during seizures (Figures 1, 2). It would be impossible for MUA sequences to be scrambled during seizures as compared to the baseline states, but for single-unit sequences between the two states to remain the same. Instead, it is possible that the MUA sequences *mask* changes in single-unit sequences. Two different single-unit sequences may have the same MUA sequence, but it is impossible for two distinct MUA sequences to have the same single unit sequence. Therefore, we feel that demonstrating that MUA is disrupted is sufficient to demonstrate that single-unit sequences are also disrupted. In this sense, MUA data serves as an adequate surrogate, and our findings in MUA are sufficient to support our conclusions.

We have added the following sentence to the revised Methods to clarify our approach:

Previous studies have shown that it is challenging to track the activity of single units during seizures, because distinct single-unit waveforms are lost during seizures²⁹. Single unit waveforms are also lost over the course of multi-day recordings. In light of this, we considered multi-unit activity (MUA), rather than single-unit activity, during seizures, consistent with previous studies^{3,20,25}. Therefore, rather than examining sequences of individual neurons, we instead examined the sequence in which anatomic locations were activated.

We have also added the following sentence to the revised Discussion to expand upon this point.

Because it is difficult to track the activity of single units during seizures²⁹, we instead analyze spatiotemporal sequences of neuronal activation by capturing the times of peak MUA at each electrode. Although we only track MUA, if sequences of MUA are disrupted, so too must be the underlying sequences of single-unit activity.

Spike bursts and the evolution of stereotypical population patterns have been described in the rat (Bragin et al. Neuroscience, 1997). It would be constructive if the authors attempted to compare their findings to animal work.

We thank the reviewer for pointing us to this valuable work. We have added the following sentence to the discussion.

Our results are consistent with findings in animal studies that suggest that excitation in hippocampal structures can entrain population spiking activity in surrounding regions (Bragin, et al. Neuroscience 1997).

The authors are addressing whether ictal bursts lead to pathological entrainment. If this is the case, it should be reflected in post-seizure baseline or interictal bursts. However, in this study, interictal epochs were separated by at least 6 hours from ictal epochs (line 390). This separation may create an artificial difference of the brain state in which IEDs and normal bursts were analyzed. It would be useful to also analyze interictal bursts and baseline bursts preceding and following seizures and demonstrate how the postictal bursts are affected.

We thank the Reviewer for this very good suggestion. We have added a new figure (now Figure S5) which explores this question. In short, we do not see a significant ramp-up or ramp-down phenomenon, in which there is pathological entrainment of pre- or post-ictal bursts, in our data. There is a possible trend towards increased distance from baseline centroid particularly among post-ictal IEDs, which could reflect residual post-ictal entrainment of IED sequences, but this trend is not significant. It is possible that this reflects a true effect, but our data may be underpowered to detect it. We intend to perform a more comprehensive analysis with a larger dataset to explore this point in the future.

We have added the following text to the revised Results section to illustrate this point:

We examined distance from the baseline centroid of baseline and IED burst in the immediate pre- and post-ictal states and did not find evidence of abnormal pre- or post-ictal entrainment (Supplementary Figure S5).

Furthermore the authors should differentiate between recording from seizure-onset-zone (SOZ) and functional cortex and report the respective recording location - this can only be inferred for the parietal participant, in whom apparently the recording site was outside the SOZ as it was not resected. If the cortex is already dysfunctional, it is difficult to argue that seizures will contribute to functional impairment by pathological entrainment.

The Reviewer makes a valid and important point. We acknowledge that our main analyses ask whether seizures cause disruptions in the underlying functional neural sequence code, but these analyses may not be valid if the underlying tissue is already dysfunctional. To address this point, it is important to take into account several considerations.

First, as the Reviewers have noted in their assessments, in general it is difficult, in light of our methods and study design, to support definitively our suggestion that entrainment of spiking activity contributes to functional impairment. The point the Reviewer raises here is, in our view, related to this larger criticism. We largely agree with this criticism, and we do not suggest that our data here provide direct evidence that such entrainment is indeed the neural basis of symptoms. We have now tempered this claim in our revised manuscript, and instead focus our study on the fact that ictal activity appears to cause entrainment of neuronal spiking into sequences with stereotyped spatiotemporal organization. Whether this is directly related to symptoms is hard to gauge from our results, but we posit that this disruption in neural coding could be an important focus for future investigations.

Second, it is also important to recognize that although we have (in most patients) resected brain regions in which these data were recorded, this does not necessarily mean that the resected tissue is entirely diseased. This is true for several reasons. The decision of where to place intracranial electrodes, and consequently the array, is not based on the specific SOZ, since this is unknown a priori. Instead, intracranial electrodes are placed in regions hypothesized to be involved in seizure onset and in areas where seizures are expected to spread. This is especially true for the anterior temporal lobe cortex, as seizures may be more likely to spread to this region than to originate in this region. Additionally, preoperative estimations of the SOZ may simply be inaccurate, since the placement of intracranial electrodes and the MEA is based on clinical hypotheses. Additionally, even after the putative SOZ has been identified, subsequent surgical resection boundaries are usually based on anatomic and functional constraints, and not only on the SOZ. For instance, the standard anterior temporal lobectomy for mesial temporal lobe seizures usually extends 3.5-4 cm from the temporal pole on the dominant hemisphere, and 4-5 cm from the temporal pole in the non-dominant hemisphere. This standard approach thus includes a regions of temporal lobe cortex that may not necessarily be directly involved in seizures.

Third, the use of intracranial electrodes (in epilepsy patients) to better understand the neural correlates of human cognition has been well established and supported by hundreds of studies. Although there is always some uncertainty regarding the epileptic brain, the consensus is that these studies have provided valuable information regarding normal human cognitive function and its underlying mechanisms. Thus, even in the case that these recordings are captured from regions that are near the SOZ, our belief is that the underlying brain still retains at least a degree of normal function, and that recordings from these brain regions are informative. Indeed, it is possible that the idea of functional and dysfunctional cortex is a false dichotomy. Instead, cortex containing the seizure source may just be the most dysfunctional, while neighboring areas to which the source spreads are less dysfunctional but still abnormal. The impact of seizures on native brain function may dissipate as distance from the seizure source increases.

Finally, it is important to point out that regardless of whether or not the cortex is functional, entrainment of baseline sequences still occurs during seizures. This may provide insight into how seizures can disrupt baseline neural activity in healthy brain regions, even if the tissue underlying the array was diseased.

Despite all of these points, we agree with the Reviewer that this is an important consideration. As such, we have added the following text to the Methods section.

Intracranial electrodes may be placed so as to record from regions directly involved in seizure onset and spread, as well as from regions which may be unaffected by seizures—consistent with standard clinical practice. Therefore, the MEA may or may not record from tissue directly involved in seizures. Subsequent surgical resection boundaries are determined by anatomic and functional constraints, as well as by the clinical team’s best estimate of seizure localization.

In the discussion (line 338) the authors propose that the entrainment of spiking sequences may serve as a biomarker for seizure onset - however they do not show any data about the seizure onset zone - see also previous point. Moreover the authors suggest that returning to baseline sequences can be a biomarker for seizure termination - they should explain better what would be the benefit of such biomarkers.

We thank the Reviewer for raising this point. When we mentioned that entrainment could serve as a biomarker for seizure onset, we meant that it could serve as a biomarker for *temporal* seizure onset, not for *spatial* seizure onset. In other words, entrainment could tell us precisely when seizures had arrived at a particular region, but would not tell us that the seizure itself had arisen strictly from that region.

We agree, however, that many markers for temporal seizure onset already exist, and that a biomarker for seizure offset might not be particularly useful, and may be achievable by simpler means. Therefore, we have removed claims that our findings may serve as a biomarker for seizure onset or offset from the discussion.

Instead, we have reframed our discussion, and the major interpretations of our findings, to claim that the entrainment we observe may disrupt neuronal function and coding. It is challenging to directly link such disruptions with symptoms, since this will depend on which brain regions are being examined and which symptoms are being evoked and/or probed. We have included this additional point in the revised Discussion:

Thus, our data suggest that traveling pathological discharges in focal epilepsy entrain sequences of spiking activity in local neural populations, thereby limiting the flexibility of neural coding. Whether such disruptions are directly related to the onset of clinical symptoms observed during seizures, however, remains unclear. Our recordings only capture the entrainment of spiking sequences in one small patch of cortex. In most cases, this is in the anterior temporal lobe, which may play a role in semantic cognition⁵⁰⁻⁵², but whose disruption may not be clinically salient. The entrainment process we observe here may in fact also transpire in other brain regions which underlie critical functions, the disruption of which may give rise to observed seizure symptoms. However, the timing relationship between entrainment in different brain regions cannot be assessed with our methods. Although our study cannot resolve the relationship between spiking entrainment and clinical symptoms definitively, the disruption of normal patterns of neural activity that we demonstrate here may nonetheless provide insight into a complementary mechanism by which seizures create symptoms.

For an audience outside the technical field it is very difficult to interpret the paper, especially figures 3 and 4. The reader needs to constantly switch between the main text and the methods section. As the findings of the paper may be interesting for non-technical readers, including clinical audience, the authors should find a way to better explain the analysis methods in the results, for example by adding cartoons to the figures, more detailed graphs labels and brief methods explanations, rather than solely referring to the methods section. The legends of the figures are very concise and often incomprehensible on their own without referring to the methods section. Also, see some examples in the minor comments, but the figure legends and labels should be generally much more informative.

We appreciate these helpful suggestions and we agree that it is important to make our results accessible to non-technical readers. We have comprehensively and substantively revised our Results section and figure legends to make them easier to follow. For instance, we have now expanded the figure legends to include brief interpretations of the results, not simply the results themselves. We have also divided the previous Figure 4 into two separate figures, now Figures 4 and 5, for clarity, and to make the contents more digestible.

In addition, we have elected to change our dimensionality reduction procedure from UMAP to PCA. The reason is that PCA is much more canonical, commonly-used, and the distances that are computed in PCA space are more easily interpreted. Our results are similar whether we use PCA versus UMAP, but we have elected to use PCA in our revised manuscript to avoid any potential questions related to how to interpret distances in UMAP space.

We hope these changes make our manuscript more accessible, but please let us know if there are still points of confusion.

minor points:

line 107 - the reference to figure S2 is wrong

Thank you for identifying this error. We meant to refer to supplementary table S2. We have addressed this in the text.

fig 1a- the scale bar annotation (sigma) is not clear for a non-technical reader and should be made more clear how sigma is related to the mean firing rate (blue traces). Also what is the scale for the red traces? (LFP).

In our initial version of the manuscript, we intended the scale bar in Figure 1 to reflect sigma for both LFP and MUA. In both cases, we were presenting z-scored data. However, we realize that may be confusing. We have now revised the scale bars in Figure 1a to just show raw units (voltage for LFP and firing rate for MUA).

In addition, based on comments on this matter from both Reviewers, we now have changed our procedure for detecting spiking bursts. We no longer use a z-score computed within each epoch to define a threshold. Instead, we now use an absolute spiking rate threshold based on the aggregate activity across all interictal epochs for each participant. We use the same threshold to identify bursts in both ictal and interictal epochs. We have described this approach in the revised Methods:

For each participant, we concatenated all interictal two-hour epochs to generate a time series of mean smoothed spike rate across all interictal epochs. We identified the spiking rate corresponding to five standard deviations above the mean of this time series. We used this absolute spiking rate as a threshold for detecting spiking bursts both during all interictal epochs as well as in all ictal epochs. We marked every time the mean smoothed spiking rate exceeded this threshold as a spiking burst event.

fig 1a - please comment on why the firing rate is highest for interictal bursts. Amplitude and time calibrations are missing.

We thank the Reviewer for raising this point. In response to comments from both Reviewers, we have now included an additional figure analyzing spike rate across seizure, IED and baseline states (Figure S2). The Reviewer correctly notes that in the exemplary bursts shown in Figure 1a, spike rate is highest for IEDs. These IEDs were associated with particularly high spike rates. However, in general, across participants, spike rate tends to be highest for seizures, intermediate for IEDs, and lowest for baselines. We have now included this new analyses in the revised Results:

Across all participants and epochs, seizure bursts were associated with the highest spike rate, while IED and baseline bursts were associated with progressively lower spike rates, respectively (Figure S2).

Are fig. 1a and fig 3a identical traces?

Yes they are. We replicated the trace from Figure 1a in Figure 3a to help orient the reader to the meaning of the spatial array configurations, shown below the traces. We have now clarified this point in the revised legend for Figure 3:

For each of the same four example seizure bursts (mean population spiking rate in dark blue trace; micro-LFP in red traces; copied from Figure 1a), we mapped the timing of peak MUA activity onto the spatial layout of the MEA.

lines 357 - 358 methods - the patient description is confusing as one of the listed participants was not included in the analysis. This participant seems irrelevant to the study and could be excluded. In summary three patients had a MEA implanted in medial temporal gyrus, one in temporal cortex but not in medial temporal gyrus (2 sessions) and one in parietal cortex. It is questionable whether the parietal recording should be pooled with the temporal recordings, especially since this patient did not have burst-related IDs (Table S3)

We agree that referencing this participant who was not included in the analysis is unnecessarily confusing. We therefore removed mention of this participant from our methods and results and supplementary tables (Supplementary Tables S1 and S2). We have also removed the previous version of Supplementary Figure S1, which illustrated a seizure from this excluded participant. For the sake of transparency, we have still noted that there was an additional participant that we had considered, but chose not to include in our analyses. We have made this point in the revised Methods:

We considered one additional participant for inclusion in our study, but in this individual, we did not observe that seizures were recruited to the MEA. We therefore did not include this participant in our data that we present here.

We also agree that it is possible that the single participant with an implant in the parietal lobe should be

considered separately from the remaining temporal lobe participants. We therefore performed several of our core analyses after removing the parietal participant. We have now reported this in the revised Results and have included a new figure (Supplementary Figure S7) that includes these results. We have added the following text to the revised Results:

We also confirmed that these differences were present when excluding the participant with a parietal MEA and focusing only on temporal lobe regions (Supplementary Figure S7).

Table S1 - several abbreviations mentioned in the legend (LGG, FCD) - do not appear in the actual table. Also some abbreviations seem to be spelled out differently in the table and legend.

We thank the Reviewer for pointing this out and we have corrected these errors.

fig 2f is missing x and y labels

Thank you for pointing this out. We have added axis labels to the figure.

fig s1a - the legend seems unrelated to the figure and it is not clear what the blue and red bars in the right plot are

We agree with the Reviewer that this figure was not clear. This figure originally presented data from a participant that was not ultimately included in our analyses. Based on the earlier comments that this participant should be not mentioned, we have elected to remove this figure altogether.

figs2 b/d - y axis - different scales or wrong numbers? the figure legend refers to r2 but the y-axis label is firing rate correlation, which is confusing

Thank you for raising this point. In the initial manuscript, the units in the y-axis label in the figure were shown as uV. This was incorrect and has been changed. However, the y-axis label is intended to reflect the cross correlation of firing rate (with units of Hz squared). For clarity, we have changed the y-axis label to “Mean electrode pairwise cross-correlation of firing rate (Hz²)”. The R² described in the legend refers to how well a Gaussian fit approximates the cross correlations we observe. In other words, R² reflects the similarity between the actual mean cross-correlation (*blue dots*), on the one hand, and the line of fit (*red curve*) on the other.

The Reviewer is also correct that in panel d, the y-axis scale is about an order of magnitude higher. The reason is that the spike rate becomes extremely high twice during the artifactual burst (panel c). This creates an extremely high value of the spike rate cross-correlation. In fact, the cross-correlation becomes very high at three points. One, at zero ms, when action potentials coincide with each other. And twice more, at around ±15 ms, when the two artifactual spike surges coincide with each other.

To clarify these points, we have added the following text to the figure legend.

In order to measure whether bursts were artifactual, we obtained the mean cross-correlation of burst rasters, across all pairs of micro-electrodes, and fitted this mean cross-correlation to a Gaussian (see Methods). The rationale was, in physiologic bursts (a), there was typically a smooth temporal gradient of action potential times, leading to a smooth mean cross-correlation that resembled a Gaussian (b). On the other hand, in many artifactual bursts, action potential times often coincided, and/or were

temporally-displaced from each other (c), leading to erratically-shaped mean cross-correlations (d). We then captured the goodness of fit, or R^2 , of the Gaussian (red curves) to the mean cross-correlation (blue dots). In other words, R^2 can be thought of as the similarity between the red curve and the blue dots.

line 438 - the authors probably mean volume conduction (instead of condition)

Thank you for pointing this out. We have corrected the error.

please explain the sleep detection algorithm as the sleep duration is unusually high for epilepsy patients (line 396 : from 6.20 ± 2.28 interictal epochs per patient on average 2.60 ± 1.14 epochs per participant were awake)

Thank you for raising this issue. The asleep clips were marked by a clinical epileptologist (S.K.I.) experienced in the review of video-iEEG. Furthermore, the epochs included in this study were not collected at random over the 24h period. Instead, the epileptologist would usually aim to select equal number of asleep and awake epochs. For some patients, more asleep epochs than awake epochs were chosen. However, again, this choice was arbitrary, and is not reflective of the underlying frequency of asleep and awake states.

We have added the following text to the methods section:

The clinical epileptologist marked awake versus asleep states based on video-iEEG review. An effort was taken to ensure that, in each participant, a roughly equal number of awake versus asleep epochs were chosen.

Reviewer #2 (Remarks to the Author):

In this study, the authors recorded intracranial EEG from depth electrodes alongside single unit activity and 'micro-scale' LFP signals from six multi-electrode arrays (MEAs) in five patients with medically refractory focal epilepsy. They recorded between 1 and 6 seizures in each patient that 'recruited' single unit activity recorded on the implanted MEA, and then compared the spiking dynamics of population burst events between these seizures, interictal epileptiform discharges (IEDs) and 'baseline' periods.

First, they find evidence for a consistent temporal structure in multi-unit activity across MEA contacts both within and between seizures – as indicated by a greater Spearman's rank correlation compared to IED and baseline bursts. Next, they use the Uniform Manifold Approximation and Projection (UMAP) method to embed each spiking sequence in a low-dimensional space, further characterise the relationship between spiking sequences in different bursts, and track their evolution over the course of each seizure. They find that seizure burst sequences rapidly diverge from baseline sequences after seizure onset and become more consistent over time as they do so. Next, they show that seizure bursts tend to exhibit greater spatial structure than random null sequences, and that this spatial structure (or 'directionality') also increases over time along with distance from the centroid of baseline sequences in UMAP space. Finally, they show that the directionality of spiking activity only matches the directionality of the LFP signal in the final third of seizures.

Overall, the manuscript is clearly presented, the data presented are rare and valuable, and the analyses appear to be sound. However, the authors make little or no attempt to link their neural data

to behaviour, and it is difficult to know what theoretical contribution is made by this study – i.e. what we can learn about epilepsy, neural coding, or cognitive function from these results. These concerns are described in more detail below.

We thank the Reviewer for these positive comments and helpful critiques. We have substantially revised our manuscript, and we feel that these revisions have substantially strengthened our findings and conclusions.

Main Concerns

I agree that temporal coding is an interesting and often overlooked phenomena in single unit recordings across species. However, from the examples shown in Figure 1, it also seems that there are clear firing rate differences between multi-unit bursts of activity occurring during different time periods of interest (seizures, IEDs, and baseline) in these data, but these differences are overlooked in the analyses and results. From a methodological perspective, it would be useful to demonstrate that differences in firing rate do not introduce a confound into the temporal sequence analyses (i.e. by down-sampling the spiking data so that firing rates are equal across bursts from different time periods of interest and then repeating the burst sequence analyses). From a theoretical perspective, it would also be useful to characterise these differences in firing rate more clearly by making formal comparisons between mean firing rates during bursts in each period, and by analysing the consistency of firing rates between bursts in the same and different periods.

We thank the Reviewer for these valuable comments. We agree that there are clear differences in firing rates between seizure, IED, and baseline bursts. We have therefore provided a new analysis in Figure S2 which examines spiking rates across states and over the course of seizure events.

In our procedures for analyzing spike sequences, we are careful to use methods which limit the effect of spike rate on our detected sequences. For instance, when comparing sequences of MUA using Spearman's rho, we consider only the *timing* of peak MUA at each electrode. We do not consider the spike *rate* at the time of peak MUA. It is true that differences in spike rate may manifest as differences in the number of missing values in each burst, which in turn could influence computations of Spearman's rho, of dimensionality reduction, and of R^2 . We address this issue as follows. When comparing two sequences using Spearman's rho, only electrodes which have MUA in both bursts are considered in the comparison, so missing values are never used. In dimensionality reduction, electrodes with missing MUA are simply infilled with random numbers prior to input into the dimensionality reduction algorithm, so missing values cannot influence the procedure in a meaningful way. Finally, in analyses involving R^2 , actual R^2 values for MUA sequences are compared head-to-head with R^2 values generated from null sequences, in which MUA times are shuffled *only* in those electrodes which are actively spiking. Therefore, the count and indices of missing values are retained in the null sequences. So, any differences in R^2 in MUA sequences, as compared to null, must arise from the actual content of the sequences, and not from the count or indices of missing values.

Despite these precautions, it is hard to completely eliminate the possibility of a confound related to spike rate. Therefore, as the Reviewer has suggested, we have now provided an additional analysis in which we downsample spike rates so that they are approximately equal across seizure, IED, and baseline states. We found that this did not materially change our results. Our findings are described in detail in a new figure (Supplementary Figure S6).

We have added these new analyses to the revised Results:

Across all participants and epochs, seizure bursts were associated with the highest spike rate, while IED and baseline bursts were associated with progressively lower spike rates, respectively (Supplementary Figure S2).

We confirmed that these differences between seizure, IED, and baseline bursts were not a consequence of the higher spike rates observed during the seizure bursts (Supplementary Figure S6).

Figure 2 could be much clearer. First, I am not sure that varying the colour code used for seizure bursts to indicate the time since the first seizure burst in panel A makes these results particularly easy to interpret. It might be preferable to first show the three different types of burst (seizure, IED, baseline) in three strongly contrasting colours, and then show the time evolution of distance from the baseline centroid in panel C only. Or perhaps, show a second version of this plot with the three second moving average of an example seizure trajectory marked in UMAP space as a thick black line with an arrowhead? Second, it might be useful to add the mean distance from baseline bursts to the baseline centroid from Fig 2b as a red dashed line onto Figure 2c, to illustrate the evolution of seizure bursts in relation to the baseline centroid more clearly. Finally, it might be worth using a separate, distinctive colour map for the Spearman's rho values in Fig 2f, to dissociate those from the colour code indicating time since the first seizure burst in all other panels

We thank the Reviewer for these excellent suggestions. We have now modified Figure 2 accordingly. We appreciate the Reviewer taking the time to provide these constructive suggestions.

There is little mechanistic interpretation or attempt to link the neural data to behaviour or clinical symptoms. In the Discussion, the authors “suggest that ... constraints on the flexibility of spike sequences may contribute to disruptions in normal cognitive processes”, but in the caption of Figure S10, they state that “clinical symptoms begin before or within seconds after the first seizure-related burst” – i.e. before burst sequences become disrupted and the flexible temporal code observed during baseline periods is constrained. These two statements seem to be completely contradictory. What we are left with is a fairly complex story about the time evolving impact of seizure activity on neural firing, with no real idea of what this means either for understanding the cognitive impact of epilepsy, seizure generation, or ‘healthy’ brain function. As such, it is difficult to identify the implication of these results, or what conclusions should be drawn from these findings.

We agree with the Reviewer that it is important to clarify the implication of these results. We have now reframed our manuscript and revised our abstract, introduction, and discussion to emphasize the main conclusion of work. Although there are several mechanisms through which seizures can disrupt neural activity, we believe our work demonstrates an additional mechanism by seizures may disrupt normal brain function—that is—by entraining neuronal activity into stereotyped and directional spiking sequences. As the Reviewer points out, it is challenging to directly relate our scientific findings to clinical symptoms, and so we have tempered these claims in our current manuscript. We removed (formerly) figure S10, because of the conceptual problems which you mentioned. Instead, we have focused our main conclusions on the findings we can support: that seizures disrupt and entrain neuronal spiking sequences. Although we present this as the main conclusion of our work, the question of how this related to clinical symptoms is still nonetheless important. We now address this as well in our revised discussion. We note that microelectrode arrays are generally placed in brain regions that are clinically-silent. Our data provide direct evidence that neural activity is disrupted in these regions. This allows us

to infer that similar disruptions *may* occur in other brain regions as well. However, we cannot assess this directly in our data. We have clarified this point in the revised Discussion.

Thus, our data suggest that traveling pathological discharges in focal epilepsy entrain sequences of spiking activity in local neural populations, thereby limiting the flexibility of neural coding. Whether such disruptions are directly related to the onset of clinical symptoms observed during seizures, however, remains unclear. Our recordings only capture the entrainment of spiking sequences in one small patch of cortex. In most cases, this is in the anterior temporal lobe, which may play a role in semantic cognition, but whose disruption may not be clinically salient. The entrainment process we observe here may in fact also transpire in other brain regions which underlie critical functions, the disruption of which may give rise to observed seizure symptoms. However, the timing relationship between entrainment in different brain regions cannot be assessed with our methods. Although our study cannot resolve the relationship between spiking entrainment and clinical symptoms definitively, the disruption of normal patterns of neural activity that we demonstrate here may nonetheless provide insight into a complementary mechanism by which seizures create symptoms.

Minor Comments

Lines 116-118: The following statement could be phrased more clearly: “Spike sequence similarity among seizure bursts was significantly greater than similarity between seizure bursts, IED bursts, and baseline bursts”, as it seems to imply that “similarity among seizure bursts was ... greater than ... similarity between seizure bursts”, which is confusing

We agree this is confusing. We have rephrased this sentence to the following.

Spike sequence similarity among seizure bursts was significantly greater than similarity in any other comparison, including comparisons within and across groups ($F(5, 20) = 23.57$, $p < .001$, repeated measures one-way ANOVA; $p < .001$, Tukey’s HSD test for multiple comparisons).

Lines 132-136: To confirm that burst sequences during seizures occupy a different part of the low-dimensional subspace than those that occur during baseline, the authors compare the UMAP distance between baseline bursts and the baseline centroid with the UMAP distance between seizure bursts and the baseline centroid. Could they also include IED burst sequences in this comparison, or provide a compelling reason why not?

We thank the Reviewer for this suggestion. We added IED burst sequences to the comparison. In addition, we have elected to change our dimensionality reduction procedure from UMAP to PCA. The reason is that PCA is much more canonical, commonly-used, and the distances that are computed in PCA space are more easily interpreted. Our results are similar whether we use PCA versus UMAP, but we have elected to use PCA in our revised manuscript to avoid any potential questions related to how to interpret distances in UMAP space.

We have included the exact comparison that the Reviewer has requested in our revised Results:

Across participants, there was a significant influence of burst state on distance from the baseline centroid ($F(2,8) = 119.701$, partial $\eta^2 = 0.97$, $p < .001$, repeated measures one-way ANOVA; Figure 2b). Seizure bursts were significantly displaced from baseline bursts and from IED bursts, although IED bursts were not significantly displaced from baseline bursts.

Line 145: Around this point, it would be useful to provide details regarding the duration of seizures across patients, to provide some context for the use of a 3s moving average

Thank you for this suggestion. At this particular place in the text, we have added the following sentences:

The average seizure duration was 243.51 ± 217.06 seconds per seizure. To capture seizure dynamics over this timescale while reducing the influence of noise, we computed the distance between seizure bursts and the baseline centroid averaged over three-second sliding windows (Figure 2c).

Lines 148/149: Admittedly, this is a facetious point, but there is no necessary reason why the maximum (minimum) distance would be greater (less) than the initial and final distances – for example, if the distance from the baseline increased or decreased monotonically over time

Thank you for pointing this out. We have changed the text to note that the maximum or minimum distance could also be equal to the initial or final distance:

Because by definition, the maximum (minimum) distances will necessarily be greater (less) than or equal to the initial and final distances, we compared these measures of distance to the distances expected by chance.

Lines 197-200: Can the authors test whether seizure bursts exhibited greater spatial structure (i.e. were better fit by a plane) than bursts that occur during IEDs or baseline periods? I think this would be particularly relevant to the interpretation of these findings

We thank the Reviewer for raising this interesting question which we had not considered directly in the initial manuscript. We have added this analysis (now Figure 3b, c). There is a significant difference between seizure and null sequences and IED and null. While we do not see significance between all comparisons, the data suggest that there is a step-wise decline in R^2 from seizure, to IED, to baseline, to null. We have added the following to the text.

In this example seizure, the mean directionality, R^2 , across both seizure and IED bursts was greater than the mean R^2 for random null samples obtained by scrambling the ordering of spiking in seizure bursts (Figure 3b). We repeated this analysis for all participants and arrays. Across participants and arrays, there was a significant effect of state on R^2 ($F(3,12) = 8.72$, partial $\eta^2 = 0.69$, $p = .002$, repeated measures one-way ANOVA; Figure 3c). There was a statistically-significant difference between seizure and null R^2 and between IED and null R^2 .

Lines 278/279: “As the seizure evolves, spiking sequences diverge from the sequences observed ... during interictal discharges” – I am not sure that this result is actually shown anywhere in the manuscript, the authors only show that spiking sequences diverge from baseline. Please edit accordingly

We agree that this is also a good point that is related to the previous comment. As the Reviewer has suggested, we have added IEDs to the comparison of distance to baseline centroid, now in Figure 2b, which provides support for this claim.

Methods, multi-unit activity burst detection: It seems concerning that different criteria are used to identify bursts during different periods, which are then compared as like for like in the main manuscript. Can the authors demonstrate that the main results are qualitatively unaffected by using consistent burst detection criteria?

We agree with the Reviewer that it may be problematic to use different z-score thresholds for burst detection in different states. We have therefore revised our approach towards burst detection. Now, we simply concatenate all interictal epochs prior to burst detection. A time series of mean smoothed spiking rate is then retrieved over all concatenated epochs. We designate the burst detection threshold as the firing rate which is 5 standard deviations above the mean based on the entirety of this time series. This same raw firing rate (not z-score) threshold is then applied across both interictal and ictal epochs. We found that the results are nearly identical under this scheme, but it is simpler and less prone to unintended consequences or biases.

We added the following text to the manuscript.

For each participant, we concatenated all interictal two-hour epochs to generate a time series of mean smoothed spike rate across all interictal epochs. We identified the spiking rate corresponding to five standard deviations above the mean of this time series. We used this absolute spiking rate as a threshold for detecting spiking bursts both during all interictal epochs as well as in all ictal epochs. We marked every time the mean smoothed spiking rate exceeded this threshold as a spiking burst event.

Thank you again for considering our manuscript. We look forward to your reply and to the reviews of our manuscript.

Sincerely,

Kareem A. Zaghoul, MD, PhD
Surgical Neurology Branch, NINDS
National Institutes of Health
Building 10, Room 3D20
10 Center Drive
Bethesda, MD 20892-1414
O: (301) 594-8114
F: (301) 402-0380
kareem.zaghoul@nih.gov

REVIEWERS' COMMENTS

Reviewer #1 (Remarks to the Author):

The authors responded effectively to all my critical points by adding new analyses and explaining the phenomena they describe in more detail. They also tempered several of their previous claims.

Reviewer #2 (Remarks to the Author):

The authors have done much to improve this manuscript, and I am now happy to recommend it for publication. I suggest some final minor changes to the text below, which I think will improve the clarity of the paper

Abstract: The authors begin by stating that "Seizures produce clinical symptoms by disrupting neural coding", but perhaps (a) they should specify that they are referring to epileptic seizures and (b) be less definitive, given that they acknowledge in the Introduction that there is no strong evidence to support this assertion

Results, new text: I think that "Sequence similarity was also weak in comparisons between burst states" would be clearer than "Sequence similarity was weak in comparisons across burst states"

Also that "per seizure" could be removed from the sentence "The average seizure duration was 243.51 ± 217.06 seconds per seizure"

Also that the authors should specify whether there was a statistically significant difference between R^2 values for baseline bursts and the null distribution, where they state that: "There was a statistically-significant difference between seizure and null R^2 and between IED and null R^2 "

In the discussion, I think it would be worth briefly commenting on the apparent discrepancy between the stability of burst patterns between the middle and final third of seizures (i.e. as shown in Figure 2C) and the change in angular direction of burst patterns between the middle and final

third of seizures (i.e. as shown in Figure Y). These results would appear to contradict one another, to some extent

Figure 2H is still labelled as 'UMAP'

REVIEWERS' COMMENTS

Reviewer #1 (Remarks to the Author):

The authors responded effectively to all my critical points by adding new analyses and explaining the phenomena they describe in more detail. They also tempered several of their previous claims.

We sincerely thank the reviewer for the positive feedback, and for the thorough, insightful and fair review.

Reviewer #2 (Remarks to the Author):

The authors have done much to improve this manuscript, and I am now happy to recommend it for publication. I suggest some final minor changes to the text below, which I think will improve the clarity of the paper

We sincerely thank the reviewer for the positive feedback, and for the thorough, insightful and fair review.

Abstract: The authors begin by stating that "Seizures produce clinical symptoms by disrupting neural coding", but perhaps (a) they should specify that they are referring to epileptic seizures and (b) be less definitive, given that they acknowledge in the Introduction that there is no strong evidence to support this assertion

The beginning of the abstract has been changed to:

Epileptic seizures are debilitating because of the clinical symptoms they produce. These symptoms, in turn, may stem directly from disruptions in neural coding.

Results, new text: I think that "Sequence similarity was also weak in comparisons between burst states" would be clearer than "Sequence similarity was weak in comparisons across burst states"

We have made this change.

Also that "per seizure" could be removed from the sentence "The average seizure duration was 243.51 ± 217.06 seconds per seizure"

To eliminate the redundancy, we have changed it to:

The average event duration was 243.51 ± 217.06 seconds per seizure.

Also that the authors should specify whether there was a statistically significant difference between R^2 values for baseline bursts and the null distribution, where they state that: "There was a statistically-significant difference between seizure and null R^2 and between IED and null R^2 "

The sentence now reads:

There was a statistically-significant difference between seizure and null R^2 ($p = .002$) and between IED and null R^2 ($p = .02$), but not between baseline and null R^2 ($p = .13$).

In the discussion, I think it would be worth briefly commenting on the apparent discrepancy between the stability of burst patterns between the middle and final third of seizures (i.e. as shown in Figure 2C) and the change in angular direction of burst patterns between the middle and final third of seizures (i.e. as shown in Figure Y). These results would appear to contradict one another, to some extent

We thank the reader for catching this subtle but important possible inconsistency. We have thought a lot about it ourselves. In short, we think that the results are not actually contradictory. The possible discrepancy lies in the fact that entrainment does indeed begin early and abruptly after seizure onset (Figure 2c, 3b). Indeed, directionality emerges shortly after seizure onset as well (Figure 4d), and R^2 values tend to correlate with distance to baseline centroid (Figure 4e). So, it is precisely the change in MUA angular direction to match LFP that often happens late in the seizure, even though directionality itself arises early.

First, it's worth noting that there is actually a weak anti-correlation between *measures of entrainment* (distance from baseline, self-similarity, and R^2), on the one hand, and LFP to MUA angular difference, on the other: see Figure S9, bottom row. However, these anti-correlations do not reach statistical significance, and so there is in fact a timing discrepancy between these two events.

What explains this discrepancy? We believe that the change in directionality is simply the end result of a sustained period of entrainment. Note in Figure 6 (formerly 5) that the MUA spatial maps actually look fairly similar in panels a, c, and e. In 6a and 6c, the "warmest" area of these maps is just below and to the right of the center of the array, while in 5e, the "warmest" area is just above and to the left of the center of the array. So, while the angular difference distribution changes abruptly (d vs f), the spatial map itself only changes gradually. Therefore, the rapid change in the angular difference distribution stems partially from the fact that MUA direction is a "winner take all" phenomenon. Even if the spatial map changes only slightly, for instance from just left of center to just right of center, the angular direction changes drastically, from left to right.

Therefore, the evolution of directionality is actually an ongoing and gradual process that starts early in the seizure, but may only complete itself by late seizure. Why does MUA direction sometimes start opposite to LFP direction (for instance, 6b)? We still do not have a good answer for this, and the relationship between MUA and LFP is complex. This may be a topic of future study. One explanation is that particularly early in the seizure, the MUA sequences retain features of the original neural network (see Figure S10). Initial MUA direction may simply be arbitrary, and need not be opposite. For instance, in some cases, even early MUA direction matches LFP (see current Figure 7a, formerly 5h, leftmost panel). In sum, early in the seizure, the process of angular

evolution begins. The initial direction may simply be arbitrary, and may be opposite the LFP. But the end direction should match the direction of the LFP, once the entrainment process has completed itself.

We have added the following sentences to the discussion.

Abruptly after seizure onset, spike sequence consistency and stereotypy emerge (Figure 2), along with spike sequence directionality (Figure 3). Over the course of the seizure, often later in the event, spike sequence directionality evolves to match the direction of the traveling waves (Figure 6, Figure 7). The time discrepancy between sequence stereotypy, on the one hand, and directional evolution, on the other, may simply reflect the fact that the latter is the end result of the former (Figure 8).

Figure 2H is still labelled as 'UMAP'

We thank the reviewer for catching this error and have fixed it.